# Nsp1 proteins of human coronaviruses HCoV-OC43 and SARS-CoV2 inhibit stress granule formation

Stacia M. Dolliver[1], Mariel Kleer[2,3], Maxwell P. Bui-Marinos[2,3], Shan Ying[1], Jennifer A. Corcoran[2,3], Denys A. Khaperskyy[1]*

1 Department of Microbiology & Immunology, Dalhousie University, Halifax, Canada, 2 Department of Microbiology, Immunology and Infectious Diseases, University of Calgary, Calgary, Canada, 3 Snyder Institute for Chronic Diseases and Charbonneau Institute for Cancer Research, University of Calgary, Calgary, Canada

* d.khaperskyy@dal.ca

**Data Availability Statement:** All data underlying our findings is contained in the manuscript and Supporting information files.

## Abstract

Stress granules (SGs) are cytoplasmic condensates that often form as part of the cellular antiviral response. Despite the growing interest in understanding the interplay between SGs and other biological condensates and viral replication, the role of SG formation during coronavirus infection remains poorly understood. Several proteins from different coronaviruses have been shown to suppress SG formation upon overexpression, but there are only a handful of studies analyzing SG formation in coronavirus-infected cells. To better understand SG inhibition by coronaviruses, we analyzed SG formation during infection with the human common cold coronavirus OC43 (HCoV-OC43) and the pandemic SARS-CoV2. We did not observe SG induction in infected cells and both viruses inhibited eukaryotic translation initiation factor 2α (eIF2α) phosphorylation and SG formation induced by exogenous stress. Furthermore, in SARS-CoV2 infected cells we observed a sharp decrease in the levels of SG-nucleating protein G3BP1. Ectopic overexpression of nucleocapsid (N) and nonstructural protein 1 (Nsp1) from both HCoV-OC43 and SARS-CoV2 inhibited SG formation. The Nsp1 proteins of both viruses inhibited arsenite-induced eIF2α phosphorylation, and the Nsp1 of SARS-CoV2 alone was sufficient to cause a decrease in G3BP1 levels. This phenotype was dependent on the depletion of cytoplasmic mRNA mediated by Nsp1 and associated with nuclear accumulation of the SG-nucleating protein TIAR. To test the role of G3BP1 in coronavirus replication, we infected cells overexpressing EGFP-tagged G3BP1 with HCoV-OC43 and observed a significant decrease in virus replication compared to control cells expressing EGFP. The antiviral role of G3BP1 and the existence of multiple SG suppression mechanisms that are conserved between HCoV-OC43 and SARS-CoV2 suggest that SG formation may represent an important antiviral host defense that coronaviruses target to ensure efficient replication.

**Funding:** This work was supported by Canadian Institutes for Health Research (CIHR) Project Grant PJT-175130 and Research Nova Scotia Grant RNS-NHIG-2020-1383 (to D.K.). This study was also supported in part by operating funds awarded to JAC from the Canadian Institutes for Health Research (CIHR): a COVID rapid response operating grant (#177704) and an operating grant (#175622) the Coronavirus Variants Rapid Response Network (CoVaRR-Net), of which JAC is a member. MK was supported by a Cumming School of Medicine graduate training award, a Canadian Institutes for Health Research (CIHR) CGS-M scholarship, and a CIHR doctoral award. The funders had no role in study design, data collection and analysis, decision to publish, or preparation of the manuscript.

**Competing interests:** Authors hereby declare that there are no financial conflicts of interest with regards to this work.

## Author summary

Host cells possess many mechanisms that can detect viral infections and trigger defense programs to suppress viral replication and spread. One of such antiviral mechanisms is the formation of stress granules–large aggregates of RNA and proteins that sequester viral components and cellular factors needed by the virus to replicate. Because of this threat, viruses evolved specific mechanisms that prevent stress granule formation. Understanding these mechanisms can reveal potential targets for therapies that would disable viral inhibition of stress granules and render cells resistant to infection. In this study, we analyzed inhibition of stress granules by two human coronaviruses: the common cold coronavirus OC43 and the pandemic SARS-CoV2. We have demonstrated that these viruses employ at least two proteins–nucleocapsid protein (N) and the non-structural protein 1 (Nsp1) to suppress stress granules. These proteins act through distinct but complementary mechanisms to ensure successful virus replication. Because both OC43 and SARS-CoV2 each dedicate more than one gene product to inhibit stress granule formation, our work suggests that viral disarming of stress granule responses is central for productive infection.

## Introduction

Coronaviruses are a family of human and animal enveloped viruses with positive-sense RNA genomes. In humans, coronaviruses predominantly cause respiratory infections of varied severity. Four circulating seasonal common cold coronaviruses (HCoVs) belong to *Alphacoronavirus* and *Betacoronavirus* genera and include HCoV-NL63, HCoV-229E, HCoV-HKU1, and HCoV-OC43 [1]. In the last two decades, three novel Betacoronaviruses have entered human circulation from zoonotic sources and caused infections with high morbidity and mortality. This group includes the severe acute respiratory syndrome coronavirus (SARS-CoV), the Middle East respiratory syndrome coronavirus (MERS-CoV), and the SARS-CoV2 virus that appeared in late 2019 and caused the most devastating respiratory virus pandemic since the 1918 Spanish Flu [2–5].

The relatively large genomes of Betacoronaviruses are ~30 kb and encode 4–5 structural proteins and 16 non-structural proteins (Nsp1-16) [1,6–8]. The non-structural proteins are synthesized from capped and polyadenylated genomic RNA as a polyprotein encoded by a large open reading frame (ORF), ORF1ab. This polyprotein is proteolytically processed into mature proteins by two viral enzymes: Nsp3 papain-like proteinase (PLpro) and Nsp5 3C-like proteinase (3CLpro) [1,9]. Non-structural proteins include factors that enable viral replication in the host cell and the subunits of the RNA-dependent RNA polymerase [6]. Structural proteins are encoded by a nested set of subgenomic mRNAs produced by the viral polymerase. These subgenomic mRNAs also encode a variable number of smaller accessory ORFs depending on the virus species [8]. Structural proteins include membrane (M), envelope (E), nucleocapsid (N), and spike (S). Of these, M and E are responsible for virus particle formation, N is an RNA-binding protein that packages viral genomes into virions, and S is the receptor binding glycoprotein that protrudes from the virion envelope and mediates membrane fusion and viral entry [10]. HCoV-OC43 (OC43) S binds sialic acid that is abundant on the surface of most cell types [11], while SARS-CoV (SARS) and SARS-CoV2 (CoV2) S bind angiotensin converting enzyme 2 (ACE2), determining viral tropism for the ACE2-expressing cells [12,13].

Once the virus enters a host cell, its genome associates with host translation machinery to initiate synthesis of viral proteins involved in subgenomic mRNA production, genome replication, and subversion of intrinsic host antiviral responses [14]. Cytoplasmic replication of

coronaviruses can generate double-stranded RNA (dsRNA) intermediates that are important pathogen-associated molecular patterns (PAMPs) sensed by the host cell [14,15]. Coronavirus dsRNA is believed to be shielded in membrane bound replication transcription compartments to limit activation of cytosolic sensors [1,8]. This mechanism, however, is insufficient to fully prevent viral dsRNA sensing, because when other mechanisms of suppression of intrinsic antiviral responses are inactivated, antiviral responses are induced in coronavirus-infected cells [16,17]. Retinoic acid inducible gene I (RIG-I) and melanoma differentiation-associated protein 5 (MDA5) can detect viral RNAs and signal to transcriptionally induce antiviral cytokines such as type I and type III interferons [15,18,19]. In addition, viral replication intermediates can be recognized in the cytosol by the dsRNA activated protein kinase (PKR)—one of the four kinases that can trigger inhibition of protein synthesis by phosphorylation of the α subunit of the eukaryotic translation initiation factor 2 (eIF2α) [20]. When eIF2α is phosphorylated, it stably associates with the guanine exchange factor eIF2B, preventing regeneration of translation initiation-competent GTP-bound form of eIF2 [21]. This blocks translation initiation and, since viruses rely on host translation for their protein synthesis, it can block viral replication. In addition, inhibition of translation initiation can induce formation of stress granules (SGs) [22–24].

SGs are large cytoplasmic condensates that accumulate translationally inactive messenger ribonucleoprotein complexes and dozens of proteins and other molecules [23,24]. SG condensation is driven by SG-nucleating proteins like the Ras-GTPase-activating protein SH3-domain-binding protein 1 (G3BP1), G3BP2, T-cell internal antigen 1 (TIA-1), and T-cell internal antigen related protein (TIAR) [23–26]. In addition to these and other SG-nucleating proteins, SG condensates accumulate mRNAs and translation initiation factors. In virus-infected cells, SGs may sequester viral RNAs and proteins to either disrupt their functions in the virus replication cycle or to influence detection of viral RNA by PAMP sensors (e.g. RIG-I) [27–31]. Thus, accumulating evidence indicates that SGs are antiviral, and many viruses have evolved dedicated mechanisms that inhibit their formation [32].

Several coronavirus gene products have been reported to inhibit SG formation. Of these, the most well characterized is the N protein of CoV2 that upon ectopic overexpression, directly binds the G3BP1 protein and blocks SG formation induced by a variety of exogenous stressors [33–36]. Proteomic analyses have also identified other interactors of N that are components of SGs (e.g. cytoplasmic poly(A) binding protein 1, PABP) as well as α and β subunits of the casein kinase 2 that can phosphorylate G3BP1 and promote SG disassembly [36,37]. In addition, Nsp15 of infectious bronchitis virus (IBV) was shown to inhibit SG formation [17,38]. Nsp15 is a nuclease that is conserved in coronaviruses; it preferentially cleaves polyuridine RNA sequences and inhibits accumulation and detection of viral dsRNA in infected cells [39]. The nuclease activity of Nsp15 was shown to be important for SG inhibition by IBV as well as the suppression of interferon (IFN) production. The SG suppression function of Nsp15 was shown to be conserved in other CoVs, as overexpression of Nsp15 from transmissible gastroenteritis virus (TGEV), porcine epidemic diarrhea virus (PEDV), SARS, or CoV2 were all shown to suppress SG formation [17]. In addition to N and Nsp15, the 4a protein from MERS-CoV can block SGs [40]. Binding of dsRNA products by 4a limits activation of PKR and downstream translation inhibition and SG formation [40].

Another viral protein that was shown to affect SG formation is Nsp1, a host shutoff factor encoded by the first N-terminal portion of ORF1ab of *Betacoronaviruses* [41]. The Nsp1-mediated host shutoff plays a key role in suppressing innate immune responses in infected cells, which has been demonstrated by comparing wild-type and recombinant Nsp1 mutant CoV infections with transmissible gastroenteritis virus (TGEV) [42], mouse hepatitis virus (MHV) [43], MERS-CoV [44], SARS [45], and CoV2 [46]. Nsp1 functions primarily through

inhibition of host protein synthesis; for SARS and CoV2 Nsp1, this is accomplished through binding to the 43S small ribosomal subunit complex and blocking the mRNA entry channel of the mature 80S ribosome [41,47,48]. In addition, Nsp1 proteins of SARS and CoV2 induce host mRNA degradation by a yet to be identified host nuclease [49–51]. Nsp1 of MERS-CoV has also been reported to degrade cellular mRNAs [52,53]. MERS-CoV Nsp1 predominantly targets mRNAs encoding ribosomal protein genes, resulting in a lack of active ribosomes and decreased global translation in infected cells [52]. Upon ectopic overexpression, SARS Nsp1 was shown to interact with G3BP1 and modify the composition of sodium arsenite (As) induced SGs by diminishing G3BP1 recruitment [54]. G3BP1 and other SG nucleating proteins drive SG formation by interacting with polysome-free messenger ribonucleoproteins that accumulate after inhibition of translation initiation. Therefore, we envision two possible complementary mechanisms by which Nsp1 may influence SG dynamics. Inhibition of translation initiation by Nsp1 may promote SG formation by causing polysome disassembly. At the same time, host mRNA degradation may deplete untranslated mRNAs and inhibit SG formation, consistent with our prior work showing that influenza A virus host shutoff endonuclease PA-X potently inhibits SG formation through this mechanism [31]. Which of the two Nsp1 host shutoff features dominates the SG response to human CoV infection and how Nsp1 proteins that do not induce mRNA degradation modulate SG formation has not been tested to date.

Given the proposed antiviral functions of SG formation and the multiple potential mechanisms that Betacoronaviruses employ to block SG formation and modify their composition, we sought to characterize SG inhibition mechanisms by the model common cold Betacoronavirus OC43. We compared SG inhibition by OC43 to the pandemic CoV2 in the same cell culture infection model to identify potential common strategies and/or differences in the magnitude and molecular mechanisms of SG inhibition. In this work we demonstrate that both OC43 and CoV2 efficiently inhibit SG formation in infected cells and show that N and Nsp1 proteins of both viruses act through distinct mechanisms to inhibit SG formation. N proteins of OC43 and CoV2 act independent from eIF2α phosphorylation and downstream of translation arrest, while Nsp1 proteins block SG formation by inhibiting eIF2α phosphorylation upstream of SG nucleation. In addition, CoV2 but not OC43 infection causes depletion of G3BP1 and disrupts nucleocytoplasmic shuttling of TIAR, which contributes to more potent inhibition of SG formation by this virus. We demonstrate that the CoV2 Nsp1-mediated mRNA degradation function is responsible, at least in part, for depletion of G3BP1 and nuclear accumulation of TIAR. Overexpression of G3BP1 significantly decreased OC43 replication, illustrating that G3BP1 is antiviral towards coronaviruses. Our study reveals that OC43 and CoV2 each dedicate more than one gene product to limit SG formation, a genetic redundancy that supports that viral disarming of SG responses is central for a productive infection.

## Results

### Human coronavirus OC43 inhibits SG formation in infected cells

Previously in our laboratory we established a robust OC43 infection model in human embryonic kidney (HEK) 293A cells [55]. This cell line, historically used for isolation and titration of adenoviruses [56–58], is also readily infected with many RNA viruses, including OC43, which rapidly replicates in 293A cells to high titers. To examine SG dynamics over the course of infection, we infected 293A cells with OC43 at multiplicity of infection (MOI) of 1.0 and analysed infected cells for the presence of SGs at various times post-infection using immunofluorescence staining for TIAR protein (Fig 1A). We observed little to no SG formation until 48 h post infection (hpi), by which time assessing SG formation became difficult because many cells started lifting off and dying from infection. On average, only 5% of infected cells formed SGs

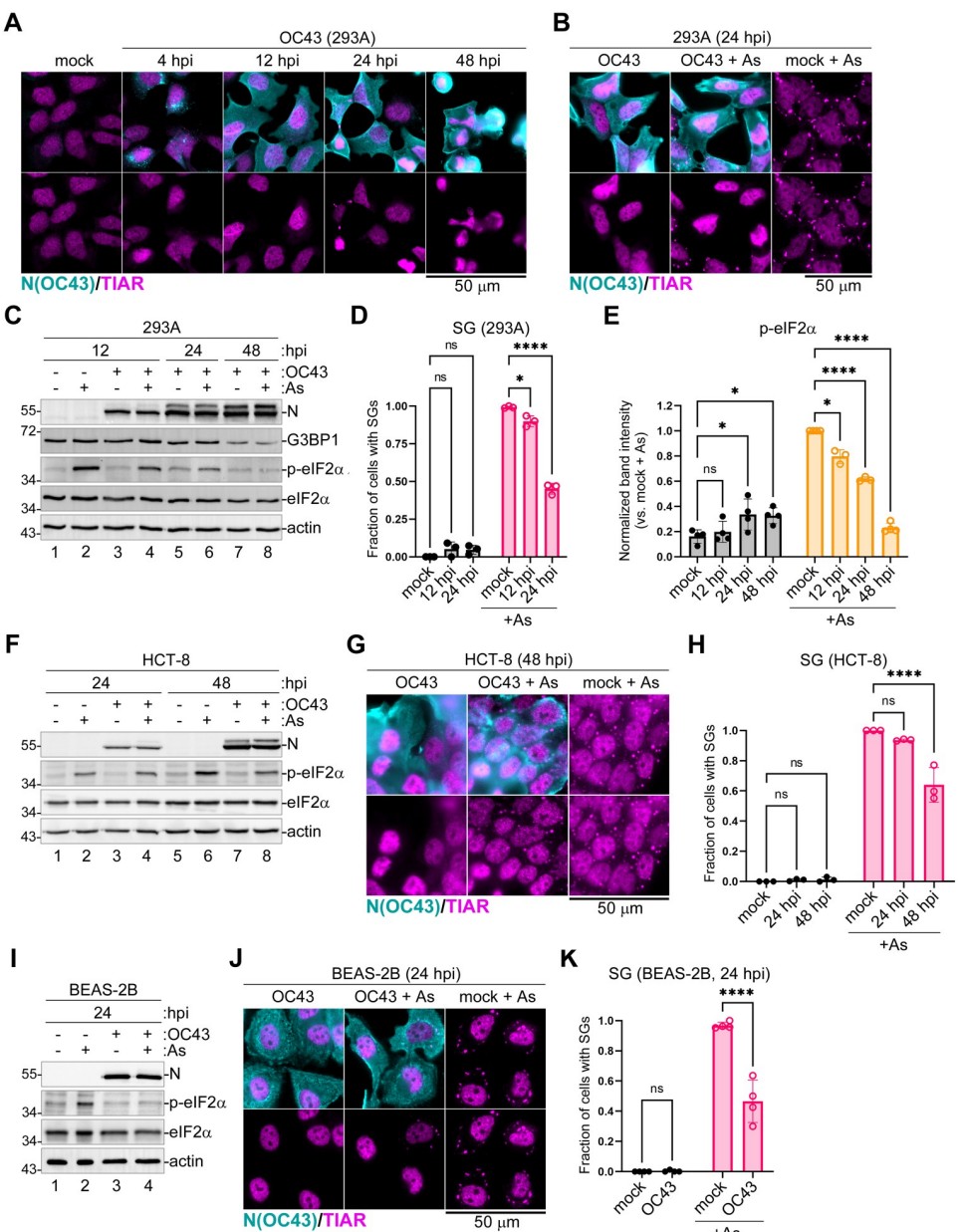

**Fig 1. Coronavirus OC43 inhibits SG formation and eIF2α phosphorylation.** Cells were infected with OC43 and SG formation in was analyzed at the indicated times post-infection using immunofluorescence staining for nucleoprotein (N(OC43), teal) and SG marker TIAR (magenta). Levels of N protein accumulation and phosphorylation of eIF2α were analysed by western blot. hpi = hours post-infection. Scale bars = 50 μm. (A) Immunofluorescence analysis of infected 293A cells at different times post-infection. (B,G,J) Immunofluorescence analysis of SG formation in mock infected and OC43-infected 293A (B), HCT-8 (G), and BEAS-2B (J) cells treated with sodium arsenite (+ As) or untreated infected cells at indicated times post-infection. (C,F,I) Western blot analysis of As-induced eIF2α phosphorylation and accumulation of N protein in 293A (C), HCT-8 (F), and BEAS-2B (I) cells at the indicated times post-infection. Levels of SG nucleating protein G3BP1 were also analyzed in (C). Actin was used as loading control. (E) Levels of eIF2α phosphorylation were quantified in 293A cells from western blot analyses as presented in (C) and values normalized to total eIF2α band intensities were plotted. (D,H,K) Fraction of cells with SGs was quantified in mock and OC43-infected 293A cells at the indicated times post-infection. Two-way ANOVA and Tukey multiple comparisons tests were done to determine statistical significance (*, p -value < 0.05; ****, p -value < 0.0001; ns = non-significant). On all plots each data point represents independent biological replicate (N ≥ 3). Error bars = standard deviation.

at 24 hpi (Fig 1A and 1D). Because there was little SG formation in OC43-infected cells, we analysed if OC43 actively inhibited SG formation. We treated mock and virus-infected cells with sodium arsenite (As). As is a potent SG-inducing agent which is commonly used to induce high levels of eIF2α phosphorylation and SG formation. It causes oxidative stress and activates eIF2α kinase heme-regulated inhibitor (HRI) [22,25]. As expected, in mock-infected cells, SGs were induced in nearly 100% of cells. By contrast, less than half of the OC43-infected cells formed SGs following As treatment at 24 hpi (Fig 1B and 1D). Next, we tested levels of eIF2α phosphorylation in infected cells. Our analysis revealed that OC43 infection inhibited As-induced eIF2α phosphorylation with increasing efficiency from 12 to 48 hpi, and the increasing magnitude of phospho-eIF2α inhibition correlated with inhibition of As-induced SG formation (Fig 1C–1E). Mechanistically, decrease in SGs can result from impaired assembly and/or accelerated disassembly. In our infection model, OC43 affected the rate of SG assembly and did not increase the rate of SG dissolution or levels of SG disassembly-promoting factors such as chaperones HSP90α/β or the dual specificity kinase Dyrk3 [59] (S1 Fig). Thus, OC43 blocks SG condensation rather than turnover, and this phenotype, at least in part, is due to viral inhibition of eIF2α phosphorylation-induced translation arrest upstream of SG nucleation.

To verify that our observations are not specific for 293A cells, we repeated analyses of OC43 effects on As-induced eIF2α phosphorylation and SG formation in human colon (HCT-8) cells, which are often used to grow this virus, and the immortalized primary human upper airway epithelial BEAS-2B cells that more closely represent a cell type infected by coronaviruses *in vivo*. As-induced eIF2α phosphorylation and SG formation were inhibited in HCT-8 cells, however at the later time point, 48 hpi (Fig 1F–1H), possibly reflecting slower virus replication kinetics in this cell type compared to 293A cells. Indeed, at 24 hpi, much lower levels of OC43 N protein were detected in HCT-8 cells (compare lanes 3 and 4 to 7 and 8 in Fig 1F). Similar to 293A, in infected BEAS-2B cells, eIF2α phosphorylation and SG formation were inhibited at 24 hpi (Fig 1I–1K), indicating that these phenotypes are not limited to fully transformed cell types and that our 293A infection model is appropriate for analysis of SG responses to infection.

## Inhibition of SG formation in OC43-infected cells does not depend on blocking eIF2α phosphorylation

Given that OC43 infection simultaneously decreased As-induced SG formation and eIF2α phosphorylation, we decided to test if this virus could block SG formation induced by an eIF2α phosphorylation-independent pathway. To induce SGs in these experiments, we used Silvestrol (Sil.). Sil. inhibits the helicase eIF4A, an important translation initiation factor, and triggers SG formation without inducing phosphorylation of eIF2α [60,61]. The 293A cells were infected with OC43 and treated with Sil. for 1 hour prior to analysis at 24 h post-infection. Similar to As, Sil. treatment triggered SG formation in nearly 100% of mock-infected cells, while only half of OC43-infected cells had SGs (Fig 2A and 2B), and, on average, fewer SGs of smaller size formed in those cells compared to mock (Fig 2C). As expected, Sil. treatment did not induce eIF2α phosphorylation (Fig 2D). Interestingly, in many infected cells we noticed brighter nuclear TIAR staining compared to uninfected cells, possibly indicating disruption of normal nucleocytoplasmic shuttling of TIAR (Figs 1A and 1B and 2A). To confirm that OC43 effects were not limited to TIAR-containing SGs, we completed a series of experiments using multiple SG marker proteins to analyze SG formation in infected cells treated with either As or Sil. These analyses confirmed that regardless of the markers used, including SG-nucleating proteins G3BP1, G3BP2, and TIA-1, as well as translation initiation factors eIF4G and eIF3B,

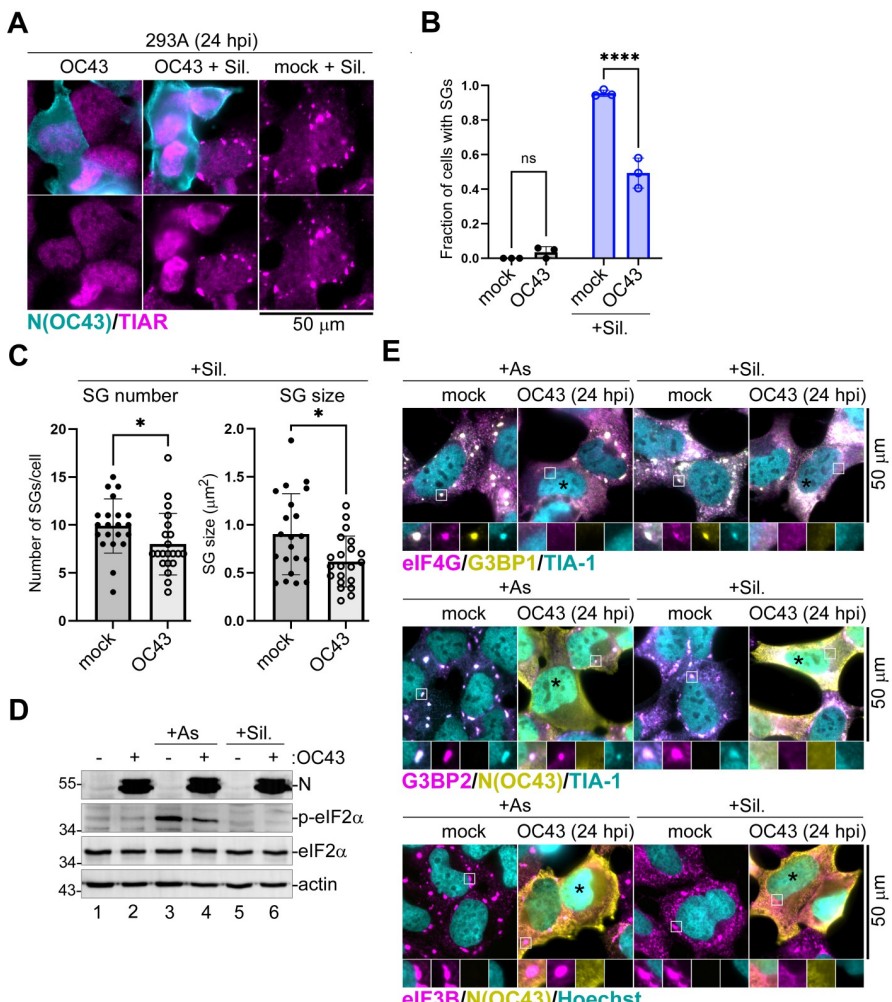

**Fig 2. OC43 inhibits SGs independently of eIF2α phosphorylation.** 293A cells were infected with OC43 at MOI = 1.0 and SG formation in arsenite (+ As), Silvestrol (+ Sil.), and untreated mock and OC43 infected cells was analyzed at 24 hours post-infection (hpi) using immunofluorescence staining for nucleoprotein and the indicated SG markers. (A) Immunofluorescence analysis of SG formation in mock infected and OC43-infected cells treated with Silvestrol (+ Sil.) or untreated infected cells. (B) Fraction of cells with SGs was quantified in mock and OC43-infected 293A cells treated and stained as in panel (A). Each data point represents independent biological replicate (N = 3). Error bars = standard deviation. Two-way ANOVA and Tukey multiple comparisons tests were done to determine statistical significance (****, p -value < 0.0001, ns = non-significant). (C) TIAR-positive SG number per cell and average SG size per cell were quantified in mock and OC43-infected 293A cells treated and stained as in panel (A). Each data point represents individual cell analysed from 3 independent biological replicates (21 cells per condition). Error bars = standard deviation. Two-tailed Students t-Test was done to determine statistical significance. (*, p -value < 0.05). (D) Phosphorylation of eIF2α was analysed by western blot. (E) Representative images of mock infected and OC43 infected cells immunostained for SG markers eIF4G (magenta), G3BP1 (yellow), TIA-1 (teal), G3BP2 (magenta), and eIF3B (magenta) as indicated. Subcellular distribution of nucleoprotein (N(OC43), yellow) was visualised by immunostaining and nuclear DNA was visualised with Hoechst dye (teal) where indicated. Black asterisks indicate infected cells that did not form SGs. Outsets show enlarged areas of cytoplasm with separation of channels to better visualize co-localization of SGs markers. Scale bars = 50 μm.

formation of SG foci was inhibited by OC43 (Fig 2E). Inhibition of Sil.-induced SGs indicates that decreased eIF2α phosphorylation in infected cells is not the only mechanism of SG inhibition by OC43 and that the virus also interferes with SG assembly downstream of translation arrest. In addition, these analyses revealed that in a fraction of infected cells that did form SGs, the OC43 N protein was not accumulating in these foci (Fig 2E).

## SARS-CoV2 blocks SG formation in infected cells and depletes SG nucleating protein G3BP1

Ectopic overexpression studies suggest that several gene products of pandemic CoV2 virus can suppresses SG formation [17,35,62,63]. To test if CoV2 can effectively block SG formation in our cell culture system, we infected 293A cells stably expressing ACE2 (293A-ACE2) with this virus and analysed SG formation. In our *in vitro* infection model, no SG formation was observed in CoV2-infected cells at either 12 hpi or 24 hpi (Fig 3A). In addition, CoV2 was able to effectively suppress SG formation induced by As (Fig 3B and 3C). Compared to OC43, CoV2 infection resulted in much greater SG inhibition, with only about 8% of infected cells forming SGs following As treatment (Fig 3C). Similar to OC43, CoV2 inhibited As-induced eIF2α phosphorylation (Fig 3D). Interestingly, we also observed an increase in nuclear TIAR signal in CoV2 infected cells, which was even more prominent than our observation in OC43 infected cells (Fig 3A and 3B). When we compared total levels of TIAR in mock and CoV2 infected cells using western blot, we observed similar levels, indicating that the virus causes changes in subcellular distribution of TIAR without drastically affecting its expression (Fig 3D and 3E). By contrast, we consistently observed substantial reduction in the levels of G3BP1 protein in CoV2 infected cells compared to mock (Fig 3D and 3E). Unlike OC43 infection, which resulted in a detectable decrease in G3BP1 levels only at 48 hpi (Fig 1C), the decrease in G3BP1 levels was observed much earlier in CoV2 infected cells (Fig 3D and 3E). To examine if these changes in G3BP1 expression were due to a decrease in its transcript levels, we isolated total RNA from mock and CoV2-infected cells at 24 hpi and analysed G3BP1 and TIAR mRNA levels using RT-QPCR. Consistent with a known feature of CoV2 host shutoff causing cytoplasmic mRNA degradation, we detected dramatic depletion of both G3BP1 and TIAR transcripts in infected cells (Fig 3F). The magnitude of mRNA depletion was slightly higher for G3BP1 (on average 80% depleted for G3BP1 vs. 60% for TIAR), but alone it would not account for the observed differences in protein levels in infected cells. To examine relative stability of G3BP1 and TIAR proteins in 293A cells, we treated uninfected cells with two translation inhibitors, cycloheximide (CHX) and Sil., as well as the transcription inhibitor Actinomycin D (ActD) for 12 hours and measured protein levels using western blot. As expected, within 1 hour of treatment, CHX and Sil. potently blocked protein synthesis in 293A cells, while ActD had no effect (Fig 3G). However, after 12 h, only ActD treatment resulted in a small (~25%) but statistically significant decrease in G3BP1 and, to a lesser extent, TIAR protein levels, while translation inhibitors did not decrease either of these proteins (Fig 3H–3J). This suggests that neither G3BP1 nor TIAR are intrinsically unstable and rapidly degraded proteins. Instead, it points to a distinct turnover mechanism for these RNA-binding proteins that is activated in response to either decrease in transcription or total mRNA levels. As a control, we examined total levels of the translation initiation factor eIF4G and detected no changes in its expression after ActD treatment (Fig 3K). These results suggest that the dramatic decrease in G3BP1 levels in CoV2 infected cells is primarily due to mRNA depletion rather than translation arrest, which are both features of CoV2 host shutoff. However, a direct targeting of G3BP1 for degradation by a viral protein cannot be ruled out. In either case, given the central role of G3BP1 in nucleating SGs, its depletion undoubtedly contributes to potent SG suppression by CoV2.

## N proteins of OC43 and CoV2 inhibit SG formation

Previous studies demonstrated that CoV2 N and Nsp15 proteins inhibit SG formation upon ectopic overexpression [17,33,36,63,64]. To test if OC43 N and Nsp15 proteins contribute to SG inhibition by this virus, we overexpressed EGFP-tagged OC43 and CoV2 N and Nsp15

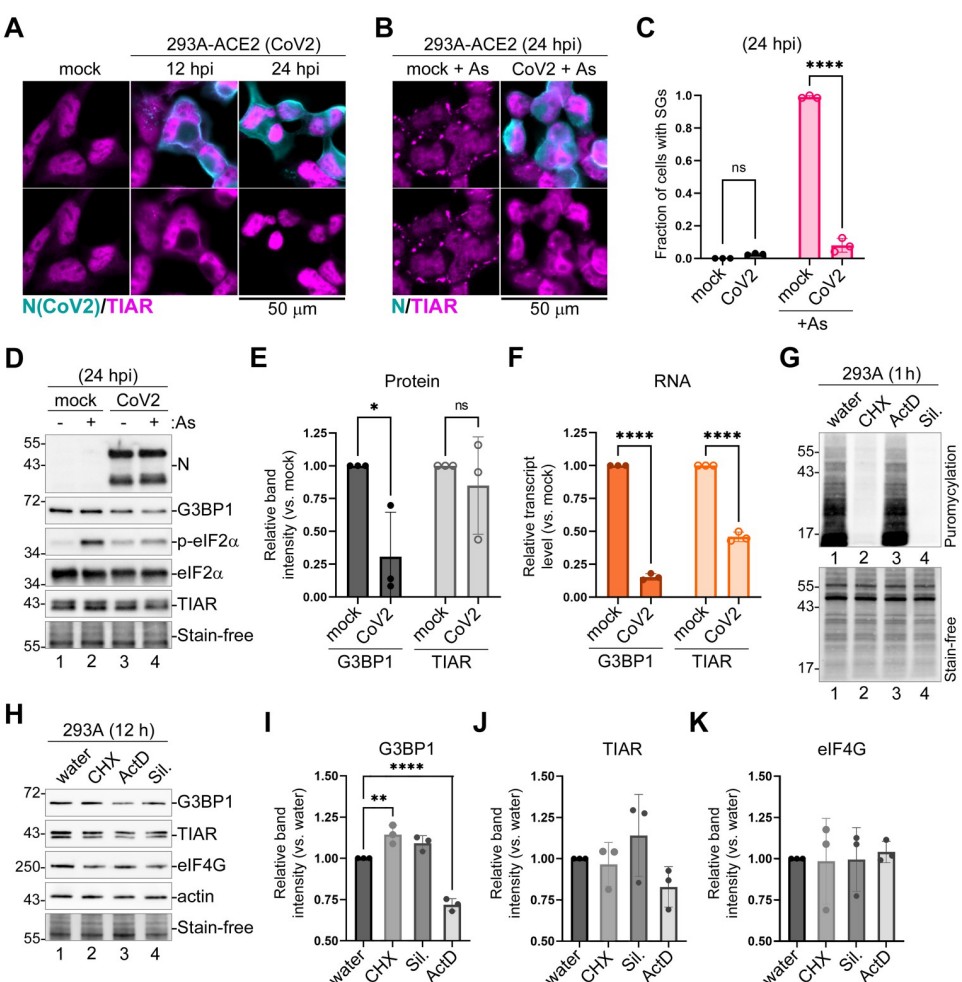

**Fig 3. SARS-CoV-2 inhibits SG formation and decreases expression of G3BP1.** (A) 293A-ACE2 cells were infected with CoV2 at MOI = 1.0 and mock and CoV2 infected cells were analysed by immunofluorescence microscopy using immunostaining for viral nucleoprotein (N, teal) and SG marker TIAR (magenta) at the indicated times post-infection. (B) Immunofluorescence microscopy of mock and CoV2 infected 293A-ACE2 cells treated with As at 24 hpi. Staining was performed as in (A). In (A) and (B) scale bars = 50 μm. (C) Fraction of cells with SGs was quantified in mock and CoV2 infected cells treated and stained as in panel (B). Error bars = standard deviation. Two-way ANOVA and Tukey multiple comparisons tests were done to determine statistical significance (****, p -value < 0.0001, ns = non-significant). (D) Western blot analysis of As-induced eIF2α phosphorylation and accumulation of N protein in 293A-ACE2 cells at 24 hpi. Levels of SG nucleating proteins G3BP1 and TIAR were also analyzed, fluorescent total protein stain (Stain-free) was used as loading control. (E) Band intensity of G3BP1 and TIAR normalized to total protein (Stain-free) quantified from 3 independent experiments represented in D (lanes 1 and 3), (*, p -value < 0.05, ns, non-significant, as determined by unpaired Students t-Test). (F) Relative transcript levels of G3BP1 and TIAR determined by RT-QPCR (****, p -value < 0.0001, ns, non-significant, as determined by unpaired Students t-Test). (G) Ribopuromcylation assay in 293A cells treated with cycloheximide (CHX), Actinomycin D (ActD), Silvestrol (Sil.) or water control. Top—puromycin incorporation detected by western blot. Bottom—total protein visualized by Stain-free dye. (H) Western blot of 293A cells treated as in G for 12h. (I-K) Relative band intensity of G3BP1 (I), TIAR (J), and eIF4G (K) normalized to total protein (Stain-free), quantified from 3 independent experiments represented in H. Two-way ANOVA and Tukey multiple comparisons tests were done to determine statistical significance (**, p -value < 0.01; ****, p -value < 0.0001). On all plots each data point represents independent biological replicate (N = 3). Error bars = standard deviation.

proteins in 293A cells and assessed their effect on SG formation. First, we analyzed SG formation in cells transiently transfected with CoV2 and OC43 N expression constructs or EGFP control and treated with As at 24 h post-transfection. Western blot analysis revealed that the OC43 and CoV2 EGFP-N fusion proteins were expressed at similar levels, and that neither of

the N constructs affected G3BP1 expression levels or As-induced eIF2α phosphorylation (Fig 4A). As expected, immunofluorescence microscopy analyses showed that nearly 100% of the EGFP expressing cells formed SGs upon As treatment, while CoV2 N expression efficiently inhibited SG formation (Fig 4B and 4C). In OC43 N expressing cells, SG formation was also inhibited, but to a lesser extent (Fig 4C). Given that N proteins did not affect eIF2α phosphorylation, we next tested if OC43 and/or CoV2 EGFP-N constructs could inhibit Sil.-induced SG formation. Indeed, both N constructs were able to inhibit Sil.-induced SG formation (Fig 4D and 4E), and the OC43 N protein was better at inhibiting Sil.-induced SGs than the As-induced SGs (Fig 4C and 4E). This indicates that N proteins of these coronaviruses directly affect SG formation downstream of translation arrest, consistent with direct interaction with SG nucleating protein G3BP1 by CoV2 N reported by previous studies [33,63]. Interestingly, inhibition of As-induced SGs by CoV2 N was shown to be augmented by methylation at arginine-95 (R95), which is positioned in a consensus RGG motif recognized by protein arginine methyltransferases (PRMTs) [65]. Despite limited homology between the OC43 and CoV2 N proteins, we were able to identify a 30 amino acid long stretch with high identity between these two primary sequences in the vicinity of CoV2 R95 (Fig 4F). However, in the OC43 sequence the corresponding arginine is substituted with lysine and the RGG motif is absent, suggesting that OC43 N is unlikely to be methylated by PRMTs in this domain (Fig 4F). We speculate that this difference may be responsible for decreased inhibition of As-induced SGs by OC43 N compared to CoV2 N.

Next, we focused on Nsp15 to determine if it can interfere with SG formation. We transfected 293A cells with EGFP-tagged OC43 or CoV2 Nsp15 or EGFP control and treated cells with As or Sil. at 24 h post-transfection to induce SGs. Despite some SG inhibition in Nsp15-transfected cells observed in few replicates, neither OC43 nor CoV2 Nsp15 significantly affect As or Sil.-induced SG formation in our experimental model (Fig 4G and 4H). Similar to N constructs, Nsp15s from OC43 or CoV2 did not affect As-induced eIF2α phosphorylation (Fig 4I and 4J). Thus, it appears that Nsp15 proteins, at least when expressed in the absence of other viral factors, are unable to block SG formation in our experimental system.

## Nsp1-mediated host shutoff contributes to SG inhibition by coronaviruses

N proteins of both OC43 and CoV2 coronaviruses are responsible, at least in part, for inhibition of SG condensation downstream of translation arrest, without affecting levels of eIF2α phosphorylation or expression of G3BP1. However, the magnitude of SG inhibition observed in EGFP-N expressing cells was much lower than in virus-infected cells. Therefore, it is likely that additional mechanisms of SG suppression may be used by these coronaviruses. Because of the link between host mRNA degradation mediated by CoV2 host shutoff and the depletion of G3BP1, we tested if Nsp1 proteins, major host shutoff factors of OC43 and CoV2, could inhibit SG formation. Importantly, only CoV2 Nsp1 was previously shown to induce degradation of cellular mRNAs [48,51,66], while both OC43 and CoV2 Nsp1 proteins inhibit host protein expression. When we transfected 293A cells with N-terminally HA-tagged Nsp1 constructs or an EGFP control and induced SG formation with As 24 h post-transfection, we saw that both OC43 and CoV2 Nsp1 were inhibiting SG formation as determined by TIAR staining (Fig 5A). We also observed that CoV2 Nsp1 was causing depletion of cytoplasmic and an increase in nuclear TIAR signal. This suggests that Nsp1 is responsible for the disruption of nucleocytoplasmic shuttling of TIAR we saw previously in CoV2 infected cells. Western blotting analysis demonstrated that our Nsp1 constructs were inhibiting production of the co-transfected EGFP reporter, as expected, confirming their activity in blocking host gene expression. Notably, CoV2 Nsp1 had stronger effect on EGFP expression, likely because of additional activity of

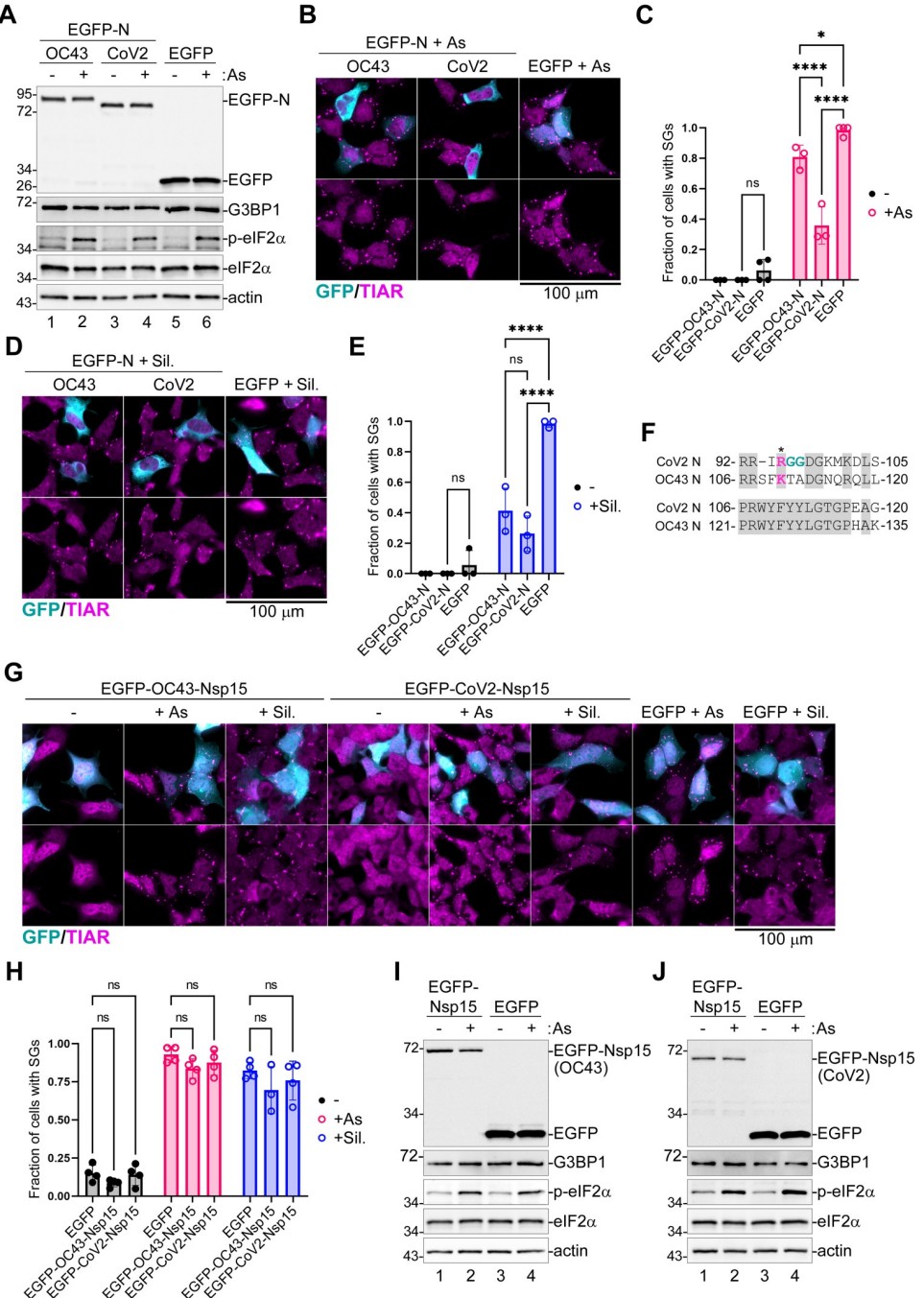

**Fig 4. Coronavirus N proteins inhibit SG formation downstream of eIF2α phosphorylation.** 293A cells were transiently transfected with the indicated EGFP-tagged viral protein expression constructs or EGFP control. At 24 h post-transfection cells were treated with sodium arsenite (+ As) or Silvestrol (+ Sil.), as indicated, or left untreated. SG formation was analyzed by immunofluorescence microscopy with staining for SG marker TIAR (magenta). EGFP expression is shown in teal. Scale bars = 100 μm. Levels of EGFP-tagged proteins and the As-induced phosphorylation of eIF2α were analyzed by western blot. (A) Western blot analysis of cells transfected with EGFP-tagged OC43 or CoV2 nucleoprotein (EGFP-N) expression vectors or control EGFP-transfected cells. (B) Immunofluorescence microscopy of EGFP-N transfected, or control EGFP-transfected cells treated with arsenite. (C) Fraction of transfected cells with As-induced SGs quantified from B. (D) Immunofluorescence microscopy of EGFP-N transfected or control EGFP-transfected cells treated with Silvestrol. (E) Fraction of transfected cells with Silvestrol-induced SGs quantified from D. (F) Alignment of CoV2 and OC43 N amino acid sequences in the vicinity of CoV2 RGG motif containing arginine 95 (red, indicated with asterisk). Identical and homologous amino acids are highlighted in grey. Numbers

indicate positions in polypeptide sequence. (G) Immunofluorescence microscopy of OC43 or CoV2 EGFP-Nsp15 transfected, or control EGFP-transfected cells treated with arsenite (+As) or Silvestrol (+Sil.). (H) Fraction of transfected cells with SGs quantified from G. (I,J) Western blot of OC43 (I) and CoV2 (J) EGFP-Nsp15 transfected or control EGFP transfected cells treated with arsenite. On all plots each data point represents independent biological replicate (N$\geq$3). Error bars = standard deviation. Two-way ANOVA and Tukey multiple comparisons tests were done to determine statistical significance (****, p -value < 0.0001, *, p-value < 0.05, ns, non-significant).

stimulating mRNA degradation (Fig 5B). In addition, western blot analysis revealed that both OC43 and CoV2 Nsp1 significantly attenuated As-induced eIF2α phosphorylation (Fig 5B and 5C). OC43 Nsp1 inhibited eIF2α phosphorylation by 30% on average, and CoV2 Nsp1 by 40%. Since we consistently observed transfection efficiencies of only 40–60% in these experiments, inhibition of p-eIF2α in transfected cells may be even stronger. Next, we tested if Nsp1 proteins affected G3BP1 protein expression. Because our HA and G3BP1 antibodies suitable for immunofluorescence staining are both mouse, we constructed N-terminally EGFP-tagged Nsp1 constructs and analysed levels and subcellular localization of G3BP1 in transfected cells treated with As. In OC43 EGFP-Nsp1 expressing cells, SG formation was inhibited and G3BP1 was diffusely distributed in the cytoplasm even upon treatment with As. In contrast, in CoV2 EGFP-Nsp1 expressing cells, G3BP1 signal was much lower than in bystander untransfected cells, EGFP expressing control cells, or OC43 EGFP-Nsp1 cells (Fig 5D). Both TIAR and G3BP1 are important SG nucleating proteins, therefore the CoV2 Nsp1 mediated redistribution of TIAR into the nucleus and depletion of G3BP1 levels could potentially disrupt cytoplasmic SG condensation. Alternatively, using TIAR or G3BP1 as SG markers in these cells could compromise visualization of SGs that may lack these proteins. To distinguish between these possibilities, we stained for G3BP2, another well established SG marker. Unlike with G3BP1 staining, we did not see significant decrease in G3BP2 signal in either OC43 or CoV2 Nsp1 expressing cells, instead we observed that some CoV2 Nsp1 expressing cells formed small G3BP2-positive SGs upon As treatment (Fig 5E). We analysed the number and size of As-induced G3BP2-positive SGs that form in CoV2 Nsp1 expressing cells and compared them to SGs formed in control EGFP expressing cells. This analysis revealed that G3BP2-positive SGs that do form in many CoV2 Nsp1-expressing cells are significantly smaller than SGs that form in control cells, while their average number remains the same (Fig 5F).

Because we observed depletion of G3BP1 protein and mRNA in CoV2 infected cells, we next tested if G3BP1 depletion by CoV2 Nsp1 but not by OC43 Nsp1 is linked to mRNA degradation induced by the former. We generated two CoV2 Nsp1 amino acid substitution mutants that are defective for mRNA degradation function but are still able to inhibit host protein synthesis: R99A N-terminal domain mutant (99A) and R124A,K125A linker region double mutant (125A) [51]. We transiently transfected 293A cells with the OC43 Nsp1, the wild type CoV2 Nsp1, with mutant CoV2 Nsp1 constructs, or with EGFP control and induced SG formation in these cells 24 h post-transfection using As treatment. Immunofluorescence microscopy analysis revealed that all Nsp1 constructs were able to inhibit As-induced SG formation to various degrees (Fig 5G). However, only the wild type CoV2 Nsp1 caused a dramatic increase in nuclear TIAR signal, while OC43 Nsp1, CoV2 99A, and 125A mutants did not affect subcellular TIAR distribution compared to EGFP control (Fig 5G). Western blot analysis of whole cell lysates revealed that none of the Nsp1 constructs altered total TIAR protein levels, indicating that, like CoV2 infection, the wild type CoV2 Nsp1 expression alters nucleocytoplasmic shuttling of TIAR without affecting its expression (Fig 5H). In addition, only the wild type CoV2 Nsp1 overexpression caused depletion of G3BP1 protein (Fig 5H, lane 3). This alteration in G3BP1 levels was not observed in OC43 Nsp1-expressing cells or cells expressing CoV2 Nsp1 mutants defective in stimulating mRNA degradation (Fig 5H, lanes 2,4,5). These

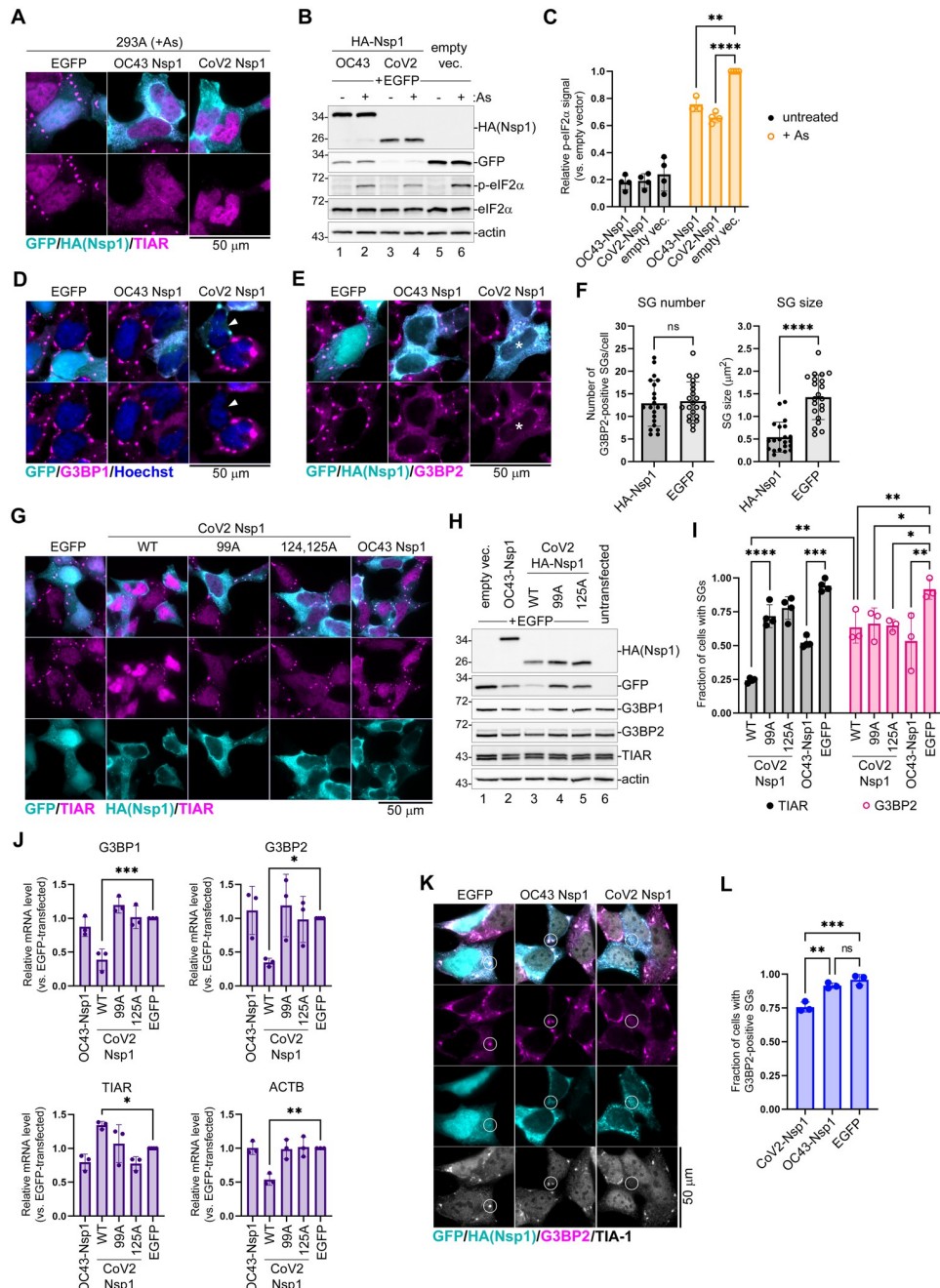

**Fig 5. Nsp1 inhibits eIF2α phosphorylation and SG formation.** (A) 293A cells were transiently transfected with the indicated HA-tagged Nsp1 expression constructs or EGFP control. At 24 h post-transfection cells were treated with sodium arsenite (+ As) and SG formation analysed by immunofluorescence microscopy staining for TIAR (magenta). GFP signal in control cells and HA-tagged Nsp1 signal are shown in teal. Scale bar = 50 μm. (B) Whole cell lysates from cells treated as in A with or without As were analysed by western blot. (C) Phospho-eIF2α band intensity normalized to total eIF2α was quantified from B and plotted relative to phospho-eIF2α level in EGFP control cells treated with As. Each data point represents independent biological replicate (N≥3). Error bars = standard deviation. Two-way ANOVA and Tukey multiple comparisons tests were done to determine statistical significance (****, p -value < 0.0001; ***, p-value <0.001; **, p-value < 0.01). (D) Immunofluorescence microscopy analysis of cells expressing the indicated EGFP-tagged Nsp1 constructs or EGFP control. SG formation was visualized by staining for G3BP1 (magenta). GFP signal is shown in teal. Arrowhead indicates a representative cell with low G3BP1 signal. Scale bar = 50 μm. (E) Immunofluorescence microscopy analysis of cells expressing the indicated HA-tagged Nsp1 constructs or EGFP control. SG formation was visualized by staining for G3BP2 (magenta). GFP signal in control cells and HA-tagged Nsp1 signal are shown in teal. Asterisks indicates a representative CoV2 Nsp1 expressing cell with

small SGs. Scale bar = 50 µm. (F) G3BP2-positive SG number per cell and average SG size per cell were quantified in CoV2 HA-Nsp1-transfected cells and control EGFP-transfected cells from E. Each data point represents individual cell analysed from 3 independent biological replicates (21 cells per condition). Error bars = standard deviation. Two-tailed Students t-Test was done to determine statistical significance. (****, p -value < 0.0001, ns, non-significant). (G) Immunofluorescence microscopy analysis of cells expressing the indicated wild type and mutant Nsp1 constructs. SG formation was visualized by staining for TIAR (magenta). GFP signal in control cells and HA-tagged Nsp1 signal is shown in teal. WT = wild type; 99A = R99A mutant; 125A = R124A,K125A mutant. Scale bar = 50 µm. (H) Western blot analysis of cells co-transfected with the indicated HA-tagged Nsp1 constructs and EGFP. (I) Fraction of transfected cells with As-induced SGs quantified from G (TIAR) and E (G3BP2). Each data point represents independent biological replicate (N≥3). Error bars = standard deviation. Two-way ANOVA and Tukey multiple comparisons tests were done to determine statistical significance (****, p -value < 0.0001; **, p-value <0.01; *, p-value < 0.05, ns, non-significant). (J) Relative G3BP1, G3BP2, TIAR, and ACTB mRNA levels were determined using RT-QPCR in cells transfected with the indicated expression constructs at 24 h post-transfections. Values were normalized to 18S. Each data point represents independent biological replicate (N = 3). Error bars = standard deviation. One-way ANOVA and Dunnett's multiple comparisons tests were done to determine statistical significance (***, p -value < 0.001; **, p-value <0.01; *, p-value < 0.05). (K) Immunofluorescence microscopy analysis of cells expressing the indicated HA-tagged Nsp1 constructs or EGFP control and treated with Silvestrol. SG formation was visualized by staining for G3BP2 (magenta) and TIA-1 (greyscale). GFP signal in control cells and HA-tagged Nsp1 signal are shown in teal. Circles highlight areas of cytoplasm with and without bright SG foci. Scale bar = 50 µm. (L) Fraction of transfected cells with silvestrol-induced SGs quantified from K (based on G3BP2 staining). Each data point represents independent biological replicate (N = 3). Error bars = standard deviation. One-way ANOVA and Tukey multiple comparisons tests were done to determine statistical significance (***, p -value < 0.001; **, p-value <0.01; ns, non-significant).

results strongly link CoV2 Nsp1-mediated host mRNA degradation to both the nuclear accumulation of TIAR and the depletion of G3BP1 protein. Indeed, when we analyzed host transcript levels in cells transfected with Nsp1 constructs by RT-QPCR, we confirmed that only the wild type CoV2 Nsp1 caused decrease in G3BP1, G3BP2, and β-actin (ACTB) mRNAs (Fig 5J). This indicated that unlike CoV2 Nsp1, the OC43 Nsp1 does not cause mRNA degradation, and that the amino acid substitution mutants we generated behave as expected. Interestingly, none of the Nsp1 constructs decreased TIAR transcript levels, with the wild type CoV2 Nsp1 causing modest but statistically significant increase in TIAR mRNA compared to EGFP control (Fig 5J).

To measure and compare SG inhibition by our panel of Nsp1 constructs, we quantified As-induced SG formation using different markers. We saw that when SGs were stained using TIAR as a marker, like in Fig 5A and 5G, the wild type CoV2 Nsp1 appeared significantly better at preventing SG formation compared to 99A or 125A mutants that did not cause nuclear accumulation of TIAR (Fig 5I). By contrast, when we quantified G3BP2-positive SGs, wild type and mutant Nsp1 constructs inhibited SG formation to the same degree (Fig 5F). While all Nsp1 constructs inhibited SG formation visualized using G3BP2 as a marker, more than 50% of cells still formed SGs. Therefore, it is apparent that in many wild type CoV2 Nsp1-expressing cells SGs still form, but they contain very little TIAR and G3BP1.

To examine if inhibition of eIF2α phosphorylation is the main mechanism of SG suppression by OC43 Nsp1, we treated OC43 Nsp1, CoV2 Nsp1, and EGFP expressing cells with Sil. and visualized SGs using G3BP2 and TIA-1 markers. Bright SG foci formed in the cytoplasm of EGFP-expressing and untransfected bystander cells, as well as in cells expressing OC43 Nsp1 (Fig 5K). At the same time, most CoV2 Nsp1 expressing cells had either smaller dispersed SG foci or no discernable SGs (Fig 5K). We quantified the fraction of cells with Sil.-induced SGs in all three conditions and demonstrated that only CoV2 Nsp1 decreased SG formation (Fig 5L). This indicates that mRNA and G3BP1 depletion, as well as nuclear retention of TIAR by CoV2 Nsp1 contribute to impaired SG condensation independent of eIF2α phosphorylation inhibition, with roughly 25% of transfected cells not forming SGs and the remaining cells forming smaller SGs. OC43 Nsp1, on the other hand, did not significantly inhibit SG formation when induced by a mechanism that is independent of eIF2α phosphorylation.

Taken together, our experiments show that both OC43 and CoV2 Nsp1 inhibit As-induced SG formation. They both act upstream by inhibiting eIF2α phosphorylation. In addition, CoV2 Nsp1 also acts downstream by affecting SG nucleation and composition. The mRNA degradation stimulated by CoV2 Nsp1 that causes nuclear accumulation of TIAR and depletion of G3BP1 protein prevents efficient SG condensation, leading to formation of smaller granules lacking TIAR and G3BP1.

## SARS-CoV2 Nsp1 inhibits PERK-mediated eIF2α phosphorylation

To further examine the mechanisms of inhibition of eIF2α phosphorylation and SG formation by CoV2 Nsp1, we created a robust system for ectopic expression of N-terminally EGFP-tagged Nsp1 in 293A cells using an inducible lentivirus vector. Previously, we have successfully used stably transduced cell lines for tightly-controlled Doxycycline (Dox)-inducible expression of host and viral proteins, including the host shutoff nuclease PA-X of the influenza A virus [67,68]. In this work, we employed the same vector backbone for creating the Dox-inducible cassette for EGFP-Nsp1 and created stably transduced 293A[iEGFP-Nsp1] cells. To induce eIF2α phosphorylation and SG formation we compared two treatments: As that causes oxidative stress and activates HRI, and the sarco/endoplasmic reticulum calcium ATPase (SERCA) inhibitor Thapsigargin (Tg) that causes endoplasmic reticulum stress and activates eIF2α kinase PERK. As expected, 500 μM As treatment triggered SG formation in nearly all uninduced cells, compared to less than 10% of cells induced for 24 h with Dox and showing cytoplasmic staining for EGFP-Nsp1. Consistent with our previous studies, 1 μM Tg induced SGs in only 25% of control cells, while almost no SGs formed in Dox-induced cells upon Tg treatment (Fig 6A and 6B). Similar to cells transfected with HA-tagged CoV2 Nsp1 expression constructs (Fig 5B and 5C), Dox-induced cells had much lower levels of As-induced eIF2α phosphorylation compared to uninduced cells (Fig 6C and 6D). Interestingly, levels of HRI kinase were lower in cells expressing EGFP-Nsp1, which alone may explain reduced eIF2α phosphorylation in response to As (Fig 6C and 6G). By contrast, EGFP-Nsp1 did not significantly affect PERK levels, but instead inhibited its phosphorylation, as indicated by the reduction of upshifted band corresponding to phospho-PERK (Fig 6E). CoV2 Nsp1 also strongly inhibited PERK-mediated eIF2α phosphorylation (Fig 6E and 6F and 6H). This clearly shows that the eIF2α phosphorylation inhibition by CoV2 Nsp1 is not specific to HRI activation induced by As and affects at least one more stress response pathway. Remarkably, the effects of Nsp1 on HRI and PERK are distinct. The HRI is downregulated, and the PERK phosphorylation is inhibited. We speculate that the HRI downregulation is the result of Nsp1-mediated host shutoff, however how Nsp1 inhibits PERK and if it affects the other two eIF2α kinases, GCN2 and PKR, warrant more focused investigation in the future.

## SARS-CoV2 Nsp1 affects subcellular distribution of TIAR and PABP

Following on our previous findings using transient transfection system, we also compared expression of SG-nucleating proteins G3BP1 and TIAR in Dox-induced and uninduced cells. As expected, induction of EGFP-Nsp1 caused significant depletion of G3BP1 (Fig 6C and 6J), while levels of TIAR were increased in Dox-induced cells compared to uninduced cells (Fig 6C and 6I). We observed an increase in TIAR mRNA levels in CoV2 Nsp1-transfected cells (Fig 5J), but we did not see increased TIAR protein levels in transfected cells (Fig 5H) or in CoV2-infected cells (Fig 3D and 3E). This suggests that TIAR expression is regulated by multiple complex mechanisms that are differentially responsive to Nsp1 depending on multiple factors (e.g. transfection or virus-induced stress, other viral proteins). Notably, in all three systems we observed increased nuclear TIAR signal (Figs 3A and 5G and 6A). This increase could result

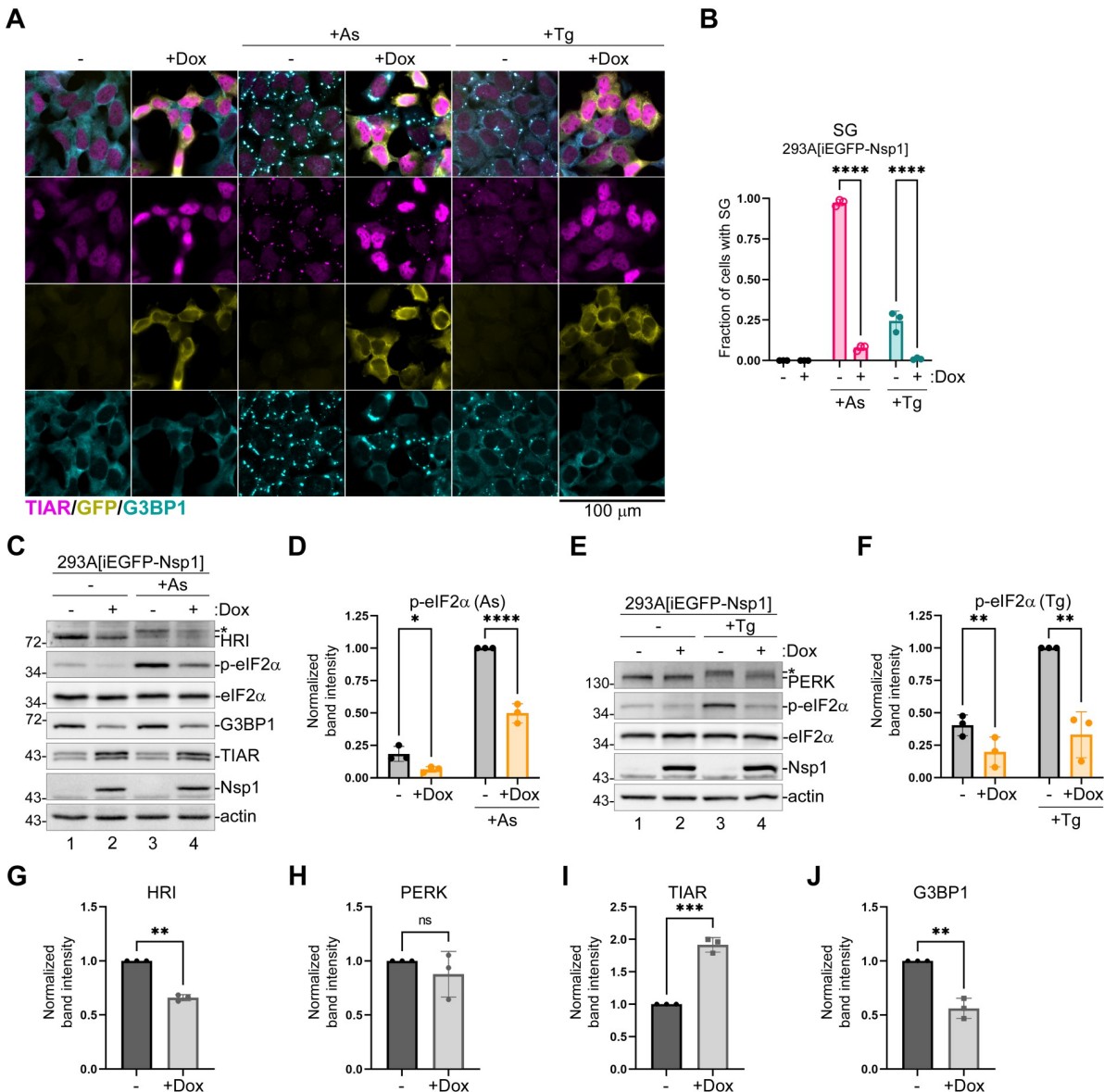

**Fig 6. CoV2 Nsp1 downregulates HRI and inhibits PERK activation.** Expression of EGFP-tagged CoV2 Nsp1 was induced in 293A [iEGFP-Nsp1] cells by treatment with doxycycline (+ Dox) for 24 h and induced and uninduced control cells (-) were treated for 50 min with 500 μM sodium arsenite (+As) or 1 h with 1 μM Thapsigargin (+Tg) where indicated. (A) Immunofluorescence microscopy analysis of SG induction using staining for TIAR (magenta) and G3BP1 (teal). GFP signal from EGFP-Nsp1 fusion protein is shown in yellow. Scale bar = 100 μm. (B) SG formation was quantified from microscopy analyses represented in A. (C) Western blot analysis of lysates from control and As-treated cells. (D) Phospho-eIF2α band intensity normalized to total eIF2α was quantified from C and plotted relative to phospho-eIF2α level in uninduced control cells treated with As. (E) Western blot analysis of lysates from control and Tg-treated cells. (F) Phospho-eIF2α band intensity normalized to total eIF2α was quantified from E and plotted relative to phospho-eIF2α level in uninduced control cells treated with Tg. (G-J) Relative band intensities of HRI (G), PERK (H), TIAR (I), and G3BP1 (J) normalized to actin were quantified from C and E. Values for untreated Dox-induced cells were plotted relative to uninduced cells. On all graphs each data point represents independent biological replicate (N = 3). Error bars = standard deviation. Two-way ANOVA and Tukey multiple comparisons tests were performed in D and E and unpaired two-tailed Student's t-Test in G-J to determine statistical significance (****, p -value < 0.0001; ***, p -value < 0.001; **, p-value <0.01; *, p-value < 0.05; ns, non-significant).

from impaired nuclear export or increased nuclear import of TIAR. It is also possible that increased levels of TIAR would result in its perceived nuclear accumulation due to smaller volume of the nucleus relative to the cytoplasm. To distinguish between these alternatives, we attempted nucleocytoplasmic fractionation of uninduced and Dox-induced 293A [iEGFP-Nsp1] cells using a modified REAP method [69]. Using this method, we achieved clear separation of nuclear and cytoplasmic fractions, as indicated by the lack of discernable traces of nuclear marker Lamin A/C in the cytoplasmic lysate and in wash fractions and absence of cytoplasmic marker Tubulin in the nuclear fraction (S2A Fig). As expected, TIAR was present in both the nuclear and the cytoplasmic fractions. Interestingly, even though nuclear TIAR increased in Dox-induced cells compared to uninduced cells, it also increased in the cytoplasmic fraction, such that the nuclear to cytoplasmic ratio did not change (S2A and S2B Fig). This result contradicts the immunofluorescence analyses that strongly suggests that the nucleocytoplasmic distribution of TIAR is affected by CoV2 Nsp1 and shows impaired recruitment of TIAR to small SGs that form in some Nsp1-expressing cells. One interpretation of this analysis would be that during fractionation, 42 and 50 kDa isoforms of TIAR detected by our antibody are close to the nuclear pore exclusion limit of approximately 60 kDa [70] and leak from the nucleus if they are not stably associated with nuclear structures. This is plausible, considering that nuclear import of TIAR is energy-dependent [71]. Therefore, to directly test if nuclear export of TIAR is impaired by Nsp1, we treated uninduced and Dox-induced 293A [iEGFP-Nsp1] cells with ActD and analyzed subcellular distribution of TIAR 2 h post-treatment. Blockade of transcription by ActD treatment was previously shown to cause egress of TIA-1 and TIAR from the nucleus [71], and in uninduced cells we observed dramatic loss of nuclear TIAR signal upon ActD treatment (S2C Fig). By contrast, in EGFP-Nsp1 positive cells, substantial nuclear TIAR staining remained even after ActD treatment (S2C Fig). This result clearly indicates that CoV2 Nsp1 at least partially inhibits nuclear export of TIAR or stimulates transcription-independent import of TIAR into the nucleus.

Given the effects of CoV2 Nsp1 on SG size and composition, we examined the small SGs that form in a fraction of CoV2 Nsp1 expressing cells upon As treatment using additional markers. Consistent with being true SGs, these granules contained translation initiation factor eIF3B (S3A Fig). These smaller granules also contained Nsp1 itself (S3A–S3C Fig). Remarkably, similar to other viral host shutoff factors that cause cytoplasmic RNA degradation [31, 72], CoV2 Nsp1 caused nuclear accumulation of cytoplasmic poly(A) binding protein (PABP), and many smaller G3BP2-positive granules lacked PABP in EGFP-Nsp1-expressing cells (S3B Fig). By contrast, these smaller SGs efficiently accumulated DDX3 and RNase L–molecules involved in innate immune responses to viruses, while chaperones HSP70 and HSP90α/β, as well as eIF2α kinase PKR were not enriched in these smaller SGs (S3C Fig). Although not exhaustive, our analysis shows that effects of CoV2 Nsp1 on SG proteins G3BP1, TIAR, and PABP results in impaired SG formation and altered SG composition. However, granules that form in Nsp1-expressing cells still contain some canonical SG markers and, like controls, do not accumulate disassembly chaperones.

## Nsp1 protein of SARS-CoV blocks SG formation and inhibits eIF2α phosphorylation

Our experiments demonstrate that despite having very low homology (Fig 7A), both OC43 and CoV2 Nsp1 proteins inhibit eIF2α phosphorylation and SG formation. To test how well these properties are conserved between Nsp1 proteins of other coronaviruses, we cloned and overexpressed N-terminally HA tagged Nsp1 proteins of human common cold Alphacoronavirus NL63 and Betacoronaviruses mouse hepatitis virus (MHV) and SARS. In untreated cells,

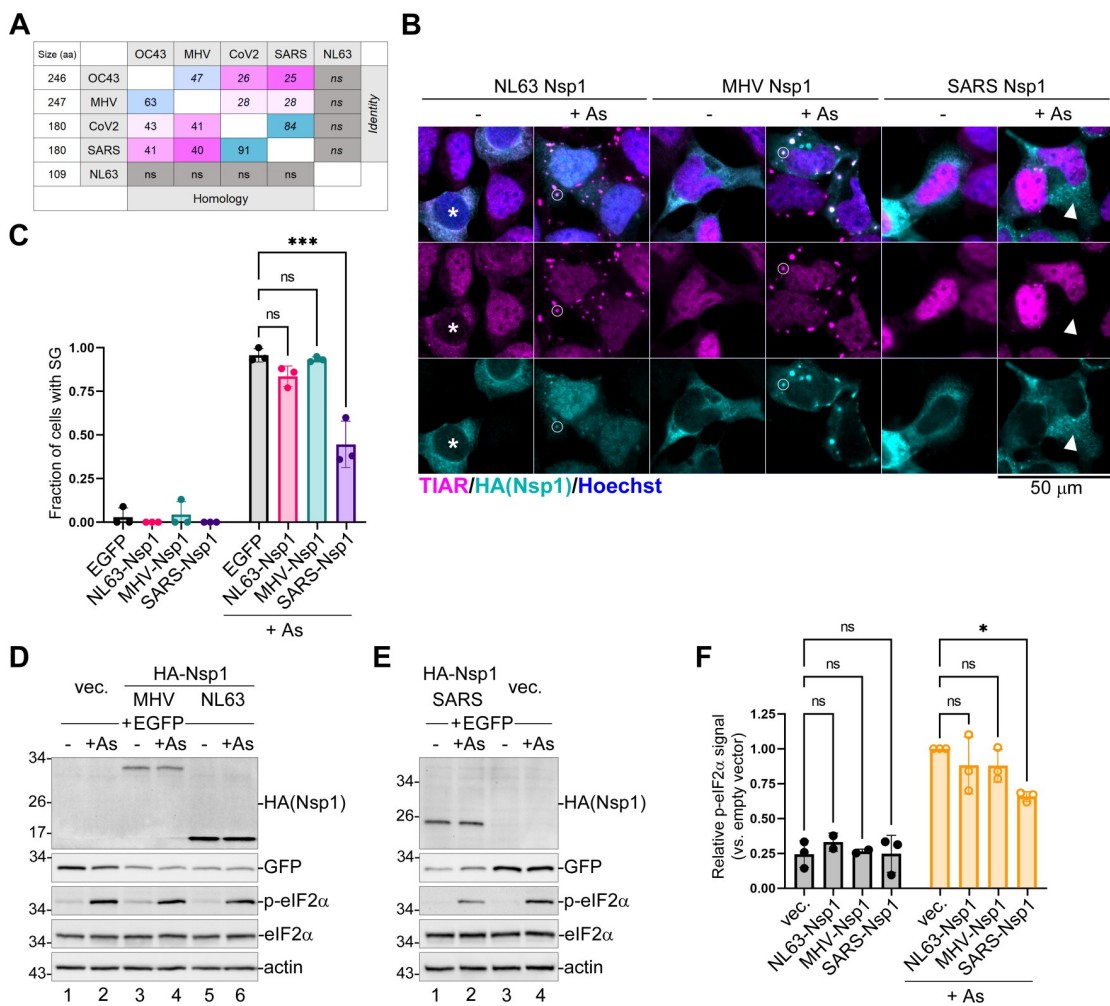

**Fig 7. Nsp1 protein of SARS CoV, but not NL63 or MHV, inhibits SG formation.** (A) Table showing percent homology (lower left) and identity (upper right) between primary amino acid (aa) sequences of Nsp1 proteins from the indicated coronavirus species as determined by pairwise NCBI BLAST alignment (https://blast.ncbi.nlm.nih.gov/Blast.cgi), ns = no significant homology. Amino acid sequences of mature Nsp1 proteins were obtained from the following accession numbers in the NCBI Protein database: YP_009555238 (OC43), NP_045299 (MHV), YP_009724389 (CoV2), YP_009944365 (SARS), and AFD98833 (NL63). (B) 293A cells were transiently transfected with the indicated HA-tagged Nsp1 expression constructs. At 24 h post-transfection cells were treated with sodium arsenite (+ As) and SG formation analysed by immunofluorescence microscopy staining for TIAR (magenta). HA-tagged Nsp1 signal is shown in teal and nuclei stained with Hoechst dye are shown in blue. Scale bar = 50 μm. Asterisk marks the NL63 Nsp1-expressing cell with depleted nuclear TIAR signal, circles highlight SGs that formed in NL-63 and MHV-expressing cells, and an arrowhead marks the SARS Nsp1-expressing cell that did not form SGs upon As treatment. (C) Fraction of transfected cells with As-induced SGs quantified from B and control EGFP-transfected cells. Each data point represents independent biological replicate (N = 3). Error bars = standard deviation. Two-way ANOVA and Tukey multiple comparisons tests were done to determine statistical significance (***, p -value < 0.001; ns, non-significant). (D,E) Whole cell lysates from cells treated as in B with or without As were analysed by western blot. (F) Phospho-eIF2α band intensity normalized to total eIF2α was quantified from D and E and plotted relative to phospho-eIF2α level in EGFP control cells treated with As. Each data point represents independent biological replicate (N = 3). Error bars = standard deviation. Two-way ANOVA and Tukey multiple comparisons tests were done to determine statistical significance (*, p-value < 0.05; ns, non-significant).

all Nsp1 proteins had cytoplasmic localization (Fig 7B) and caused decrease in expression of co-transfected EGFP reporter (Fig 7D and 7E), consistent with their function in host shutoff. When we treated transfected cells with As and analyzed SG formation and phosphorylation of eIF2α, only SARS Nsp1 significantly inhibited SG formation and decreased As-induced eIF2α phosphorylation in our system (Fig 7B–7F), while both NL63 and MHV Nsp1 proteins were

recruited to As-induced SGs (Fig 7B). SARS Nsp1 amino acid sequence has high homology (91%) to CoV2 Nsp1 (Fig 7A), and many of the previously described phenotypes associated with SARS Nsp1 were also described for CoV2 Nsp1, including induction of host mRNA cleavage and degradation [49,51]. Therefore, it was not surprising that SARS and CoV2 Nsp1 proteins would have similar effects in our system. Indeed, we also observed increase in nuclear TIAR signal in SARS Nsp1 transfected cells (Fig 7B). Interestingly, NL63 Nsp1, which is a small 109 amino acid protein with no significant homology to other tested Nsp1s (Fig 7A), had the opposite effect and caused depletion of nuclear TIAR signal, with TIAR and Nsp1 co-localizing in the cytoplasm of transfected cells. Upon As treatment, however, in most cells NL63 Nsp1 staining became predominantly nuclear and the nuclear TIAR staining also increased compared to untreated NL63 Nsp1-expressing cells (Fig 7B). To our knowledge, this is the first report of stress-induced change in nucleocytoplasmic distribution of a viral host shutoff protein.

## G3BP1 overexpression interferes with OC43 infection

G3BP1 is one of the most important SG nucleating proteins. Apart from its function in nucleating SG formation it is also involved in antiviral signaling. Since our work revealed that coronaviruses are efficient at blocking SG condensation and that CoV2 host shutoff causes G3BP1 depletion, we decided to test if G3BP1 overexpression would be detrimental to virus replication. We generated 293A cells stably transduced with lentivirus vectors encoding EGFP-tagged G3BP1 (293A[EGFP-G3BP1]) and control cells expressing EGFP (239A[EGFP]). To ensure similar levels of expression of these constructs, we sorted early passage cells to have both cell lines with similar GFP signal intensity. Although transient overexpression of G3BP1 may trigger SG formation in the absence of exogenous stress [73], we previously confirmed that this approach generated stable cell lines that did not form SGs spontaneously [30,68]. Initial testing of these cell lines revealed no major differences in cell morphology or SG formation following As treatment (S4 Fig). We infected 293A[EGFP], 293A[EGFP-G3BP1], and parental untransduced 293A cells with the same OC43 virus inoculum at the MOI = 0.1 and analyzed infection rates using immunofluorescence microscopy staining at 24 hpi. This analysis revealed that 293A[EGFP-G3BP1] cells were significantly more resistant to virus infection then either parental or control 293A[EGFP] cells (Fig 8A and 8B). Importantly, this was not due to lentiviral integration or non-specific effect of EGFP overexpression as infection rates were the same between 293A[EGFP] and parental untransduced cells (Fig 8B). Western blot analysis confirmed that the EGFP-G3BP1 fusion protein was expressed at higher levels than the endogenous G3BP1 (Fig 8C). The ectopic overexpression of EGFP-G3BP1 but not the EGFP control caused noticeable decrease in endogenous G3BP1 and G3BP2 expression, but the total level of G3BP1 still remained much higher. Consistent with lower infection rates observed in 293A [EGFP-G3BP1] cells, the accumulation of viral N protein was decreased as well (Fig 8C). To examine whether the increased resistance of 293A[EGFP-G3BP1] cells to coronavirus infection was due to increased SG formation, we analyzed infected cells at 24 hpi using immunofluorescence microscopy with staining for G3BP2 in addition to immunostaining for viral N protein. We did not observe SG formation in the majority of 293A[EGFP-G3BP1] cells, however SG formation was significantly increased compared to infected 293A[EGFP] control cells (Fig 8D and 8E). This result suggests that the OC43 virus is effective at suppressing SG formation even when G3BP1 levels are elevated. This also indicates that the antiviral function of G3BP1 is not limited to SG nucleation. To test if G3BP1 overexpressing cells are resistant to infection at early stages of virus attachment and penetration or if viral replication is inhibited by G3BP1 at later stages of infection, we conducted an infection time course in 293A

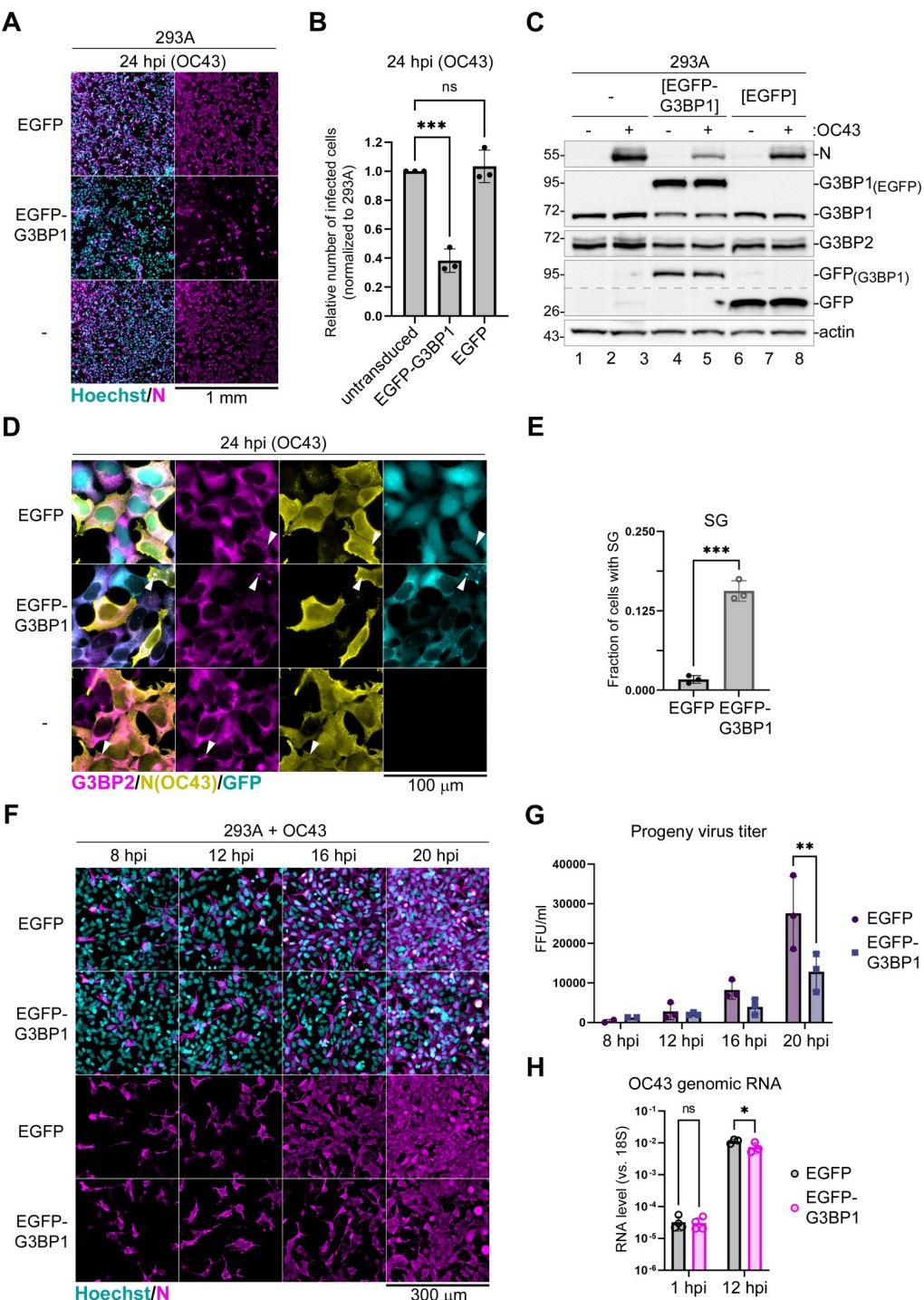

**Fig 8. G3BP1 overexpression inhibits OC43 replication.** (A) Immunofluorescence microscopy analysis of parental 293A cells (-), 293A[EGFP], and 293A[EGFP-G3BP1] cells infected with OC43 at MOI = 0.1 and stained for OC43 N protein (magenta) at 24 hpi. Nuclei were stained with Hoechst dye (teal). Scale bar = 1 mm. (B) Relative number of infected cells (normalized to 293A) in 293A[EGFP] and 293A[EGFP-G3BP1] cells at 24 hpi, quantified from A. (C) Western blot analysis of 293A, 293A[EGFP], and 293A[EGFP-G3BP1] cells infected with OC43 at 24 hpi. Anti-GFP blot was spliced along the dotted line to reduce vertical size. (D) Immunofluorescence microscopy of cells infected as in B and immunostained for G3BP2 (magenta) and OC43 N (yellow). GFP signal is shown in teal. Arrowheads indicate infected cells that formed SGs. Scale bar = 100 μm. (E) Fraction of cells with SGs was quantified in OC43-infected 293A [EGFP-G3BP1] and 293A[EGFP] cells from D. (F) Immunofluorescence microscopy analysis of 293A[EGFP] and 293A

[EGFP-G3BP1] cells infected with OC43 at MOI = 0.1 and stained for OC43 N protein (magenta) at the indicated times post-infection. Nuclei were stained with Hoechst dye (teal). Scale bar = 300 μm. (G) Infectious virus titers were determined in culture supernatants of OC43-infected cells shown in F. FFU = foci forming units. (H) Relative levels of OC43 genomic RNA were determined by RT-QPCR assay using total RNA extracted from infected cells at the indicated times post-infection. Values were normalized to 18S rRNA levels for each sample. On all graphs each data point represents independent biological replicate ($N \geq 3$). Error bars = standard deviation. One-way ANOVA and Dunnett's multiple comparisons tests (B,G,H) or the unpaired Student's t-Test (E) were done to determine statistical significance (***, p -value < 0.001; **, p -value < 0.01; *, p-value < 0.05; ns, non-significant).

[EGFP-G3BP1] and control 293A[EGFP] cells. As expected for an MOI of 0.1, roughly 10% of cells were infected in both cell lines at early 8 hpi and 12 hpi timepoints (Fig 8F). Infection rates were the same between EGFP-G3BP1 and control EGFP overexpressing cells, indicating that virus attachment and penetration was not affected by G3BP1 overexpression. At later times post-infection, we observed increasing numbers of infected cells and nearly all 293A [EGFP] cells became OC43 N positive by 20 hpi (Fig 8F). By comparison, infection spread was slower in 293A[EGFP-G3BP1] cells, with only about half the cells stained for OC43 N at 20 hpi (Fig 8F). This slower virus spread correlated with lower infectious virus production in the supernatants collected at later time points, suggesting that when G3BP1 is overexpressed, viral replication is less efficient (Fig 8G). Indeed, when we used high multiplicity of infection (MOI = 1.0) and analyzed viral genomic RNA levels as early as 1 hpi, there was no significant difference (Fig 8H). However, at 12 hpi we observed on average 2 times less viral RNA in 293A [EGFP-G3BP1] cells compared to 293A[EGFP] (Fig 8H).

## Discussion

Despite high prevalence in the population and ability to cause reinfections [74,75], common cold human coronaviruses remained poorly studied due to lower clinical significance compared to other seasonal respiratory viruses like influenza or respiratory syncytial virus (RSV). With the emergence of highly pathogenic coronaviruses of zoonotic origin, especially the recent pandemic CoV2 that swept the globe causing high morbidity and mortality, the interest in coronavirus research increased. Due to its classification as level 2 pathogen, OC43 emerged as one of the model coronaviruses [76–80]. In this study, we examined this viruses' ability to inhibit formation of SGs, one of the intrinsic host antiviral mechanisms, and compared it to that of the highly pathogenic CoV2.

Our work demonstrates that in virus-infected cells SGs do not form until about 24 h post-infection, with fewer than 5% of cells having SGs at that time point. Despite coronaviruses being known to generate levels of dsRNA sufficient to be detected by dsRNA-specific antibody [17,81], both OC43 and CoV2 do not activate PKR to levels that would induce significant eIF2α phosphorylation and SG formation. Furthermore, both viruses inhibit eIF2α phosphorylation triggered by As treatment which causes oxidative stress and activates HRI [82]. This indicates that OC43 and CoV2 suppress eIF2α phosphorylation-dependent translation arrest. The nuclease activity of Nsp15, conserved in coronaviruses from different genera, was previously shown to be important in limiting dsRNA detection in cells infected with Gammacoronavirus IBV [17]. Infection of cells with a recombinant virus with H238A substitution in Nsp15 that abrogates its nuclease activity triggered PKR activation, phosphorylation of eIF2α, and induction of SGs [17]. Thus, it is very likely that Nsp15 activity also contributes to the lack of PKR activation and SG formation in OC43 and CoV2 infected cells. Despite this, in our experimental system, overexpression of OC43 or CoV2 Nsp15 did not significantly affect As-induced eIF2α phosphorylation or SG formation. In another study, CoV2 N and the 3CLpro proteinase Nsp5 were shown to suppress SGs induced by transfection with the dsRNA mimic

polyinosinic-polycytidylic acid (poly(I:C)) [34]. However, the effects of these proteins on poly (I:C)-induced PKR activation and eIF2α phosphorylation were not examined. Previous studies have demonstrated that CoV2 N protein directly binds G3BP1 and interferes with its function in SG nucleation [33,34]. Consistent with this mechanism, upon ectopic overexpression, CoV2 N protein blocked both phospho-eIF2α dependent and independent SG formation in our study. In comparison, the OC43 N protein was also able to inhibit SGs but was better at inhibiting phospho-eIF2α independent SGs induced by Sil. than the SGs induced by As. Despite these differences in magnitude, our results show that inhibition of SG nucleation by N protein is conserved between OC43 and CoV2. Notably, neither N protein affected As-induced eIF2α phosphorylation, confirming that they function downstream at the SG nucleation step. Instead, our work revealed that OC43 and CoV2 Nsp1 host shutoff factors are responsible, at least in part, for inhibition of eIF2α phosphorylation observed in As-treated infected cells. Upon ectopic overexpression, OC43 and CoV2 Nsp1 proteins inhibited eIF2α phosphorylation and SG formation induced by As. Importantly, OC43 Nsp1 did not significantly inhibit Sil.-induced SG formation, indicating that inhibition of phospho-eIF2α mediated translation arrest is the main mechanism of SG inhibition by this protein. By contrast, CoV2 Nsp1 was also able to inhibit Sil.-induced SG formation, although to a much lesser extent compared to As-induced SGs. Since neither virus infection nor CoV2 or OC43 Nsp1 overexpression affect total eIF2α levels in our experiments, suppression of eIF2α phosphorylation by Nsp1 could be through direct inhibition of a specific kinase (e.g. HRI) or through stimulation of eIF2α dephosphorylation. When we examined eIF2α phosphorylation in cells overexpressing CoV2 Nsp1 and treated with As (activates HRI) or Tg (activates PERK), we observed that it was inhibited after either treatment. This indicates that, at least in the case of CoV2 Nsp1, the mechanism of inhibition is not specific to HRI. Activation of HRI (as judged by the relative abundance of slower-migrating HRI-specific band on the western blot) was not affected, but instead the total levels of HRI protein were decreased in Nsp1-expressing cells relative to control. By contrast, total levels of PERK were not affected by Nsp1 expression while its activation was decreased. Future studies would have to decipher the molecular mechanisms that interfere with PERK activation in cells expressing CoV2 Nsp1, as well as examine effects of Nsp1 on activation and/or expression of the other eIF2α kinases and their role in virus replication. Notably, when we examined effects of Nsp1 proteins of three other CoV species (NL63, MHV, and SARS) on eIF2α phosphorylation and SG formation in response to As, only SARS Nsp1, which is highly homologous to CoV2 Nsp1, had a significant effect. From these results, it is evident that SG inhibition function is not highly conserved between Nsp1 proteins of different CoVs and it is possible that these species have different magnitude of SG suppression or rely on other viral proteins for this function (e.g. N). It is important to note, however, that even for CoV2 and OC43, our results do not rule out the contribution of other viral proteins in the suppression of eIF2α phosphorylation in infected cells.

Another striking phenotype that we observed in CoV2 but not OC43 infected cells was the depletion of G3BP1 protein and increase in nuclear retention of TIAR. Our analysis using translation inhibitors revealed that G3BP1 protein is not intrinsically unstable, indicating that general inhibition of protein synthesis by CoV2 host shutoff is not responsible for G3BP1 depletion. Instead, we discovered that G3BP1 levels can be decreased by transcription inhibitor ActD. This suggests a link between general cytoplasmic mRNA depletion by CoV2 host shutoff factor Nsp1 and the sharp decrease in G3BP1 protein levels. Indeed, upon ectopic overexpression, CoV2 Nsp1 caused depletion of G3BP1 protein levels. By contrast, OC43 Nsp1, which does not induce mRNA degradation, did not affect G3BP1 expression. To firmly link G3BP1 protein depletion and nuclear TIAR accumulation to mRNA degradation induced by CoV2 Nsp1, we created and tested two amino acid substitution mutants that were previously shown

to be defective in stimulating host mRNA degradation–single amino acid substitution R99A in the Nsp1 N-terminal domain, and double substitution R124A,K125A in the linker region [51]. Neither mutant caused a decrease in host mRNA levels nor affected G3BP1 protein levels or nuclear TIAR accumulation, similar to OC43 Nsp1. Thus, our experiments show that G3BP1 depletion is likely a direct consequence of its mRNA degradation. At the same time, how mRNA degradation stimulated by CoV2 or SARS Nsp1 results in nuclear accumulation of TIAR is less clear. TIAR is an RNA binding protein with 3 RNA recognition motifs (RRM1-3) and an auxiliary C-terminal domain that is involved in many stages of the mRNA life cycle, from transcription and splicing to translation and stability [83]. It was shown that nuclear import of TIAR is dependent on transcription of new pre-mRNAs and requires ATP hydrolysis and the RRM2 RNA-binding activity, and that the inhibition of RNA synthesis by ActD causes egress of TIAR from the nucleus into the cytoplasm [71]. Nuclear export of TIAR requires RRM3 and its RNA binding activity, and is independent of exportin 1 (XPO1) [71]. Therefore, it is possible that TIAR bound to RNA is co-exported from the nucleus through NXF1/NXT1 dependent RNA export pathway which is inhibited by CoV2 Nsp1 [84]. However, it appears that the RNA export inhibition by CoV2 Nsp1 is independent from ribosome binding or its RNA degradation activity [46], and therefore cannot be the main driver for nuclear accumulation of TIAR. Since RNA binding by RRMs 2 and 3 has different affinity and preferred RNA sequence composition [85,86], it is possible that depletion of a certain subset of RNAs through Nsp1-induced cleavage and degradation favors RRM2-dependent nuclear import. Alternatively, Nsp1 may affect posttranslational modifications of TIAR RRMs or auxiliary domain [87,88] or disrupt its interaction with yet to be identified proteins promoting its nuclear export and SG recruitment.

Interestingly, unlike G3BP1 and actin transcripts, TIAR mRNA is not degraded in CoV2 Nsp1 expressing cells. Specific mRNA features are likely responsible for resistance of TIAR mRNAs to Nsp1-mediated degradation. For example, the first stem loop of the 5' untranslated region (UTR) of the CoV2 genome is sufficient to protect against Nsp1-mediated shutoff [89]. Transcriptomic analysis of CoV2 Nsp1 expressing cells demonstrate that a subset of host mRNAs that possess terminal oligopyrimidine tracts (TOP mRNAs) are preferentially translated and protected from degradation [90]. Furthermore, analysis of host proteins that bind to 5' UTR of the CoV2 genome identified La-related protein 1 (LARP1) and cellular nucleic acid binding protein (CNBP) [91] which are involved in regulating the translation of TOP mRNAs [92,93]. TOP mRNAs are classified as containing a capped cytidine nucleotide followed by a 7–12 nucleotide long oligopyrimidine stretch often followed by a G-rich region. TIAR is not classified as a TOP mRNA, however, it contains an extended oligopyrimidine stretch in its 5' UTR and both TIA-1 and TIAR are involved in TOP mRNA translation regulation [94]. Thus, if protection from Nsp1 degradation involves TOP mRNA machinery, this could explain why TIAR mRNA is not a target of Nsp1. Future studies should examine which sequence features of TIAR mRNA confer resistance to Nsp1.

G3BP1 protein and its homologue G3BP2 are master regulators of SG formation that can directly interact with the small ribosomal subunit and facilitate initial nucleation of SGs [26,73,95]. Consequently, most types of stress fail to induce SG formation in G3BP1/G3BP2 double knock-out cells [26,95]. Interestingly, despite partially overlapping functions, silencing of either G3BP1 or G3BP2 can inhibit both phospho-eIF2α dependent and independent SG formation, suggesting that the total levels of G3BP1/G3BP2 expression affect SG nucleation [26,68,96]. In addition, G3BP1 is involved in antiviral responses through multiple mechanisms [29,97–99]. It can amplify translation arrest by recruiting unphosphorylated PKR to stress granules, where it becomes activated in a dsRNA-independent manner [29]. It is also involved in the activation of signaling cascades leading to induction of antiviral cytokines [29,99], and

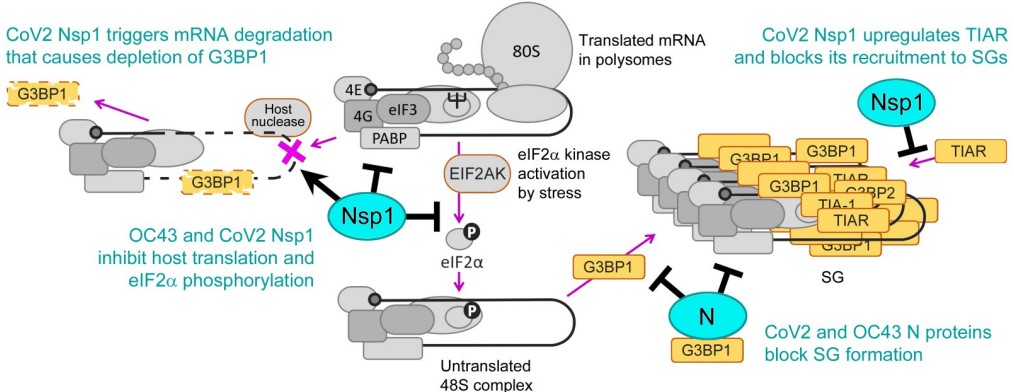

**Fig 9. Working model for the inhibition of SG formation by Nsp1 and N proteins.** Diagram illustrating concerted action of N and Nsp1 proteins in disrupting G3BP1 activity and inhibiting SG formation. Poly(A) binding protein (PABP) and the eukaryotic translation initiation factors eIF3, eIF4E (4E), and eIF4G (4G) that form preinitiation complex and recruit ribosomal subunits are shown schematically. EIF2AK = eIF2α kinase.

many studies have shown that silencing of G3BP1 leads to impaired induction of type I interferon [27,29,100]. Recently, it was discovered that G3BP1 and G3BP2 function in anchoring the tuberous sclerosis complex (TSC) to lysosomes and suppressing activation of the mechanistic target of rapamycin complex 1 (mTORC1) [101]. These functions of G3BP1 are independent from SG formation. Thus, in addition to interfering with SG nucleation, depletion of G3BP1 in CoV2 infected cells could benefit viral replication by both blunting the cellular innate immune responses and by upregulating biosynthetic pathways through relieving mTORC1 suppression. In our study we showed that overexpression of G3BP1 inhibits OC43 infection, suggesting that G3BP1 is antiviral towards OC43. Although we observed a significant increase in SG formation in infected cells overexpressing G3BP1 compared to control cells, the majority of infected cells still remained SG-free. Without ruling out the contribution of SG formation to antiviral mechanisms, this suggests that some of the SG nucleation-independent functions of G3BP1 mentioned above could play an important role in decreasing OC43 replication.

The existence of multiple mechanisms that interfere with translation arrest and SG formation in cells infected with both the seasonal common cold OC43 and the pandemic CoV2 viruses described in this study highlights the importance of overcoming these antiviral mechanisms by diverse coronaviruses. In addition, our work discovers a novel feature of Nsp1-mediated host shutoff that simultaneously blocks host translation initiation and promotes continuous regeneration of GTP-bound translation initiation-competent eIF2 by inhibiting eIF2α phosphorylation (Fig 9). In the follow up work focusing on CoV2 and OC43 Nsp1, we aim at characterizing the mechanism by which these host shutoff factors inhibit eIF2α phosphorylation and the contribution of this function to viral replication fitness and suppression of host antiviral responses.

## Materials and methods

### Cells

Human Embryonic Kidney (HEK) 293A cells and human colon adenocarcinoma (HCT-8) cells were cultured in Dulbecco's modified Eagle's medium (DMEM) supplemented with heat-inactivated 10% fetal bovine serum (FBS), and 2 mM L-glutamine (all purchased from Thermo Fisher Scientific (Thermo), Waltham, MA, USA). BEAS-2B cells were cultured in Bronchial

Epithelial Cell Growth Medium (BEGM, Lonza, Kingston, ON, Canada) on plates prepared with coating media (0.01 mg/ml fibronectin, 0.03 mg/mL bovine collagen type I, and 0.01 mg/ml bovine serum albumin (all from Millipore Sigma, Oakville, ON, Canada) dissolved in Basal Epithelial Cell Growth Medium (BEBM, Lonza)). 293A and BEAS-2B cells were purchased from American Type Culture Collection (ATCC, Manassas, VA, USA), HCT-8 cells were purchased from Millipore Sigma.

## Viruses

HCoV-OC43 was purchased from ATCC and SARS-CoV2 (strain SARS-CoV-2/SB3-TYAGNC) was derived from a clinical isolate and generously provided by Drs. Arinjay Banerjee, Karen Mossman and Samira Mubareka [102]. To generate initial HCoV-OC43 virus stocks, Vero E6 cells (ATCC) were infected at multiplicity of infection (MOI) <0.1 for 1 h in serum-free DMEM at 37˚C following replacement of the inoculum with DMEM supplemented with 1% FBS and continued incubation at 33˚C. Once CPE reached 75% at 4–5 d past-infection, the viral supernatant was harvested, centrifuged at 2,500 x g for 5 min, and then the cleared viral supernatant was aliquoted and stored at -80˚C. For SARS-CoV-2 stocks, Vero E6 cells in a confluent T-175 cm2 flask were infected at a MOI of 0.01 for 1 h at 37˚C in 2.5 mL of serum-free DMEM with intermittent shaking every 10 min. Following incubation, 17.5 ml of DMEM supplemented with 2% FBS was added directly to the viral inoculum and continued incubation at 37˚C. With the onset of CPE at 62–66 hpi, viral supernatant was harvested, centrifuged at 1,000 x g for 5 min, and then the cleared viral supernatant was aliquoted and stored at -80˚C. Stocks were titered by plaque assay on Vero E6 cells as in [103].

## Plasmids and lentivirus stocks

SARS-CoV2 and HCoV-OC43 N, Nsp1, and Nsp15 open reading frames were PCR-amplified from cDNAs generated from total RNA of infected Vero E6 cells collected at 24 hpi using specific primers with simultaneous introduction of flanking restriction sites. NL63 Nsp1 coding sequence was amplified from cDNA of NL63 virus-infected cells (kind gift from Dr. Craig McCormick, Dalhousie University), SARS Nsp1 coding sequence was amplified from pCAGGS-nsp1 vector [104], (kind gift from Dr. Marta Gaglia, University of Wisconsin-Madison), and the in vitro-synthesized MHV Nsp1 coding sequence was ordered from Invitrogen GeneArt Gene Synthesis Services (Thermo). Then, coding sequences for OC43 and CoV2 genes were inserted between EcoRI and XhoI sites into pCR3.1-EGFP vector [67] to generate pCR3.1-EGFP-OC43-N, pCR3.1-EGFP-CoV2-N, pCR3.1-EGFP-OC43-Nsp1, pCR3.1-EGFP-CoV2-Nsp1, and pCR3.1-EGFP-OC43-Nsp15 plasmids. To generate N-terminally HA-tagged Nsp1 constructs, coding sequences were inserted between KpnI and XhoI sites into 3xHA-miniTurbo-NLS_pCDNA3 vector (a gift from Alice Ting, Addgene plasmid # 107172) to generate pCDNA3-HA-OC43-Nsp1, pCDNA3-HA-CoV2-Nsp1, pCDNA3-HA-NL63-Nsp1, pCDNA3-HA-SARS-Nsp1, and pCDNA3-HA-MHV-Nsp1 vectors (miniTurbo-NLS coding sequence was replaced by Nsp1 sequences). Amino acid substitutions in pCDNA3-HA-CoV2-Nsp1 vector were introduced using Phusion PCR mutagenesis (New England Biolabs) to generate pCDNA-HA-CoV2-Nsp1(R99A) and pCDNA-HA-CoV2-Nsp1(R124A,K125A) vectors. To generate lentivirus vectors pLJM1-ACE2-BSD, pLJM1-EGFP-BSD, and pLJM1-EGFP-G3BP1-BSD, the PCR-amplified ACE2, EGFP, and G3BP1 coding sequences were inserted into the multicloning site of pLJM1-B* vector [105]. To generate pTRIPZ-EGFP-Nsp1 lentivirus vector, the CoV2-Nsp1 insert was transferred from pCR3.1-EGFP-CoV2-Nsp1 plasmid into pTRIPZ-EGFP vector [68] using EcoRI and XhoI. All constructs were verified by Sanger sequencing, sequences are available upon request.

To generate lentivirus stocks, HEK 293T cells (ATCC) were reverse-transfected with polyethylenimine (PEI, Polysciences, Warrington, PA, USA) and the following plasmids for lentiviral generation: pLJM1-B* or pTRIPZ backbone-based constructs, pMD2.G, and psPAX2. pMD2. G and psPAX2 are gifts from Didier Trono (Addgene plasmids #12259 and #12260). 48 h post-transfection, lentivirus containing supernatants were passed through a 0.45 μm filter and frozen at -80˚C.

## Generation of stably transduced cell lines

To generate 293A-ACE2 cells, 293A cells were stably transduced with a lentivirus vector encoding ACE2 (pLJM1-ACE2-BSD) and selected and maintained in 10 μg/mL Blasticidin S HCl (Thermo Fisher). To generate 293A[EGFP] and 293A[EGFP-G3BP1] cells, 293A cells were transduced with lentiviruses produced from pLJM1-EGFP-BSD and pLJM1-EGFP-G3BP1-BSD vectors and at passage 3 post-transduction, EGFP-positive cells were isolated using live cell sorting on BD FACSAria III instrument, cultured and used for experiments at passage 5 to 7. To generate 293A[iEGFP-Nsp1] cell line, 293A cells were transduced with lentivirus produced from pTRIPZ-EGFP-Nsp1 vector and selected for 48 h in the presence of 1μg/ml Puromycin, cultured and used at passage 3–6.

## Cell treatments

For SG induction, sodium arsenite (Millipore Sigma) was added to the media to a final concentration of 500 μM and cells were returned to 37˚C incubator for 50 min; Silvestrol (MedChemExpress, Monmouth Junction, NJ, USA) or Thapsigargin (Millipore Sigma) were added to the media to a final concentration of 500 nM and 1 μM, respectively, and cells were returned to 37˚C incubator for 1 h. For treatment of 293A cells with translation and transcription inhibitors, cycloheximide (50 μg/ml), Actinomycin D (5 μg/ml), or Silvestrol (320 nM) were added to the media and cells were incubated for 12 h prior to lysis for western blot.

## Virus infections

Cell monolayers were grown in 20-mm wells of 12-well cluster dishes with or without glass coverslips. For HCoV-OC43 infections, media was aspirated, cells were washed briefly with PBS and 300 μl of virus inoculum diluted to the calculated MOI = 0.1 or 1.0 in 1% FBS DMEM was added. Cells were placed at 37˚C for 1 h, with manual horizontal shaking every 10–15 minutes. Then, virus inoculum was aspirated from cells, cells were washed with PBS, 1 ml of fresh 1% FBS DMEM was added to each well, and cells were returned to 37˚C until the specified time post-infection. For SARS-CoV-2 infections, media was aspirated and 100 μL of virus inoculum diluted in serum-free DMEM at a MOI of 0.2 was added directed to the wells. Cells were incubated at 37˚C for 1h with intermittent shaking every 10 min. Following incubation, virus inoculum was removed, and cells were washed with 1 mL of PBS three times, then 1 mL of fresh DMEM supplemented with 10% FBS was added to each well. Cells were incubated at 37˚C for 24 h.

## Transfection

293A cells were seeded into 20-mm wells of 12-well cluster dishes with or without glass coverslips and the next day transfected with 500 ng DNA mixes/well containing expression vectors (250 ng) and pUC19 filler DNA (250 ng) using Fugene HD (Promega, Madison, WI, USA) according to manufacturer's protocol. Where indicated, the amount of filler DNA was reduced

to 150 ng and 100 ng of the pCR3.1-EGFP plasmid was co-transfected with expression vectors for Nsp1 proteins. Cells were used for experiments 23–24 h post-transfection as indicated.

## Immunofluorescence staining

Cell fixation and immunofluorescence staining were performed according to the procedure described in [68]. Briefly, cells grown on 18-mm round coverslips were fixed with 4% paraformaldehyde in PBS for 15 min at ambient temperature and permeabilized with cold methanol for 10 min. After 1-h blocking with 5% bovine serum albumin (BSA, BioShop, Burlington, ON, Canada) in PBS, staining was performed overnight at +4˚C with antibodies to the following targets: CoV2 N (1:400; rabbit, Novus Biologicals, NBP3-05730); DDX3 (1:200; mouse, Santa Cruz Biotechnology, sc-365768); eIF3B (1:400; rabbit, Bethyl Labs, A301761A); eIF4G (1:200; rabbit, Cell Signaling, #2498); G3BP1 (1:400; mouse, BD Transduction, 611126); G3BP2 (1:1000; rabbit, Millipore Sigma, HPA018304); HA tag (1:100; mouse, Cell Signalling, #2367); HSP70 (1:200; mouse, Santa Cruz Biotechnology, sc-32239); HSP90A/B (1:200; mouse, Santa Cruz Biotechnology, sc-13119); OC43 N (1:500; mouse, Millipore, MAB9012); RNase L (1:200; mouse, Santa Cruz Biotechnology, sc-74405); PABP (1:150; mouse, Santa Cruz Biotechnology, sc-32318); PKR (1:200; mouse, Santa Cruz Biotechnology, sc-6282); TIA-1 (1:200; goat, Santa Cruz Biotechnology, sc-1751); TIAR (1:1000; rabbit, Cell Signaling, #8509). Alexa Fluor (AF)-conjugated secondary antibodies used were: donkey anti-mouse IgG AF488 (Invitrogen, A21202), donkey anti-rabbit IgG AF555 (Invitrogen, A31572), donkey anti-goat IgG AF647 (Invitrogen, A32839). Where indicated, nuclei were stained with Hoechst 33342 dye (Invitrogen, H3570). Slides were mounted with ProLong Gold Antifade Mountant (Thermo Fisher) and imaged using Zeiss AxioImager Z2 fluorescence microscope and Zeiss ZEN 2011 software. Green, red, blue, and far-red channel colors were changed for image presentation in the color-blind safe palette without altering signal levels. Quantification of SG-positive cells was performed by counting the number of cells with at least two discrete cytoplasmic foci from at least 3 randomly selected fields of view, analysing >100 cells per treatment in each replicate. Analysis of SG number and size was performed on cropped images of individual cells using ImageJ software Analyze Particles function after automatic background subtraction and thresholding. For each of 3 independent biological replicates, 7 cells selected from at least 3 random fields of view were analyzed for a total of 21 cells per condition.

## Western blotting

Whole-cell lysates were prepared by direct lysis of PBS-washed cell monolayers with 1× Laemmli sample buffer (50 mM Tris-HCl pH 6.8, 10% glycerol, 2% SDS, 100 mM DTT, 0.005% Bromophenol Blue). Lysates were immediately placed on ice, homogenized by passing through a 21-gauge needle, and stored at −20˚C. Aliquots of lysates thawed on ice were incubated at 95˚C for 3 min, cooled on ice, separated using denaturing PAGE, transferred onto PVDF membranes using Trans Blot Turbo Transfer System with RTA Transfer Packs (Bio-Rad Laboratories, Hercules, CA, USA) according to manufacturer's protocol and analysed by immunoblotting using antibody-specific protocols. Antibodies to the following targets were used: β-actin (1:2000; HRP-conjugated, mouse, Santa Cruz Biotechnology, sc-47778); b-Tubulin (1:1000, rabbit, Cell Signaling, #2128); CoV2 N (1:1,000; rabbit, Novus Biologicals, NBP3-05730); Dyrk3 (1:1000 mouse, Santa Cruz Biotechnology, sc-390532); eIF2α (1:1000; rabbit, Cell Signaling, #5324); eIF4G (1:1000; rabbit, Cell Signaling, #2498); G3BP1 (1:4000; mouse, BD Transduction, 611126); G3BP2 (1:2500; rabbit, Millipore Sigma, HPA018304); GFP (1:1000; rabbit, Cell Signaling, #2956); HA tag (1:1000; mouse, Cell Signalling, #2367); HRI (1:1000, rabbit, MyBioSource, MBS2538144); HSP70 (1:1000; mouse, Santa Cruz

Biotechnology, sc-32239); HSP90A/B (1:1000; mouse, Santa Cruz Biotechnology, sc-13119); Lamin A/C (1:1000, mouse, Santa Cruz Biotechnology, sc-7292); OC43 N (1:1,000; mouse, Millipore, MAB9012); PERK (1:1000, rabbit, Cell Signaling, #5683); phospho-S51-eIF2α (1:1000; rabbit, Cell Signaling, #3398); TIAR (1:1000; rabbit, Cell Signaling, #8509). For band visualization, HRP-conjugated anti-rabbit IgG (Goat, Cell Signaling, #7074) or anti-mouse IgG (Horse, Cell Signaling, #7076) were used with Clarity Western ECL Substrate on the ChemiDoc Touch Imaging Sysytem (Bio-Rad Laboratories). Where indicated, total protein was visualised post-transfer to PVDF membranes on ChemiDoc using Stain-free fluorescent dye (Bio-Rad Laboratories). For analyses of protein band intensities, western blot signals were quantified using Bio-Rad Image Lab 5.2.1 software and values normalized to the Stain-free signal for each lane.

## Nucleocytoplasmic fractionation

To separate nuclear and cytoplasmic fractions of induced and uninduced 293A[iEGFP-Nsp1] cells, the REAP protocol described in [69] was followed as directed including the wash step, except the 0.1% NP40-PBS buffer was replaced with TBS lysis buffer (20 mM Tris-HCl pH 7.8, 150 mM NaCl, 1.5 mM $MgCl_2$, 0.1% Igepal, and 1 mM DTT).

## Ribopuromycylation assay

The puromycin incorporation assay was performed as described in [106] with the following modifications. Puromycin was added to the medium at the final concentration of 10 μg/ml for 10 min. Cells were washed with PBS and the whole-cell lysate preparation and western blotting analysis were done as described above. For electrophoresis, samples were loaded onto Mini-PROTEAN TGX Pre-cast Stain-Free gels (5–15%, BioRad Laboratories, Hercules, CA, USA) and total protein was visualised post-transfer to PVDF membranes on ChemiDoc Touch Imaging System. Puromycin incorporation into nascent polypeptides was visualised using anti-puromycin antibody (1:6,000; mouse, MilliporeSigma, MABE343).

## RNA isolation and RT-QPCR

Total RNA was isolated from cells using RNeasy Plus Mini kit (Qiagen) according to manufacturer's protocol. 250 ng of RNA was used to synthesize cDNA using qScript cDNA Super-Mix (Quanta) or Maxima H Minus Reverse Transcriptase (Thermo Fisher). Quantitative PCR amplification was performed using PerfeCTa SYBR Green PCR master mix (Quanta) and specific primers listed below on Cielo 3 QPCR unit (Azure). Primers used: 18S–Left: cgttcttagttggtggagcg, Right: ccggacatctaagggcatca; ACTB—Left: catccgcaaagacctgtacg, Right: cctgcttgctgatccacatc; G3BP1—Left: ggtcttaggcgtgtaccctg, Right: tatcggggaggaccctcagtg; G3BP2—Left: gcctgttaatgctgggaacac, Right: tgttgcctcctgttgcagat; TIAR—Left: tggaagatgcagaagaccgag, Right: tgcactccctagctctgaca; OC43-Nsp15—Left: atggcgtagtggtggacaag, Right: actcc-caggctgtcgaattg. Relative target levels were determined using ΔΔCt method with normalization to 18S.

## Statistical analyses

All numerical values are plotted as means (bar graphs) and display individual datapoints representing independent biological replicates (separate experiments performed on different days); the error bars represent standard deviations. Statistical analyses for each data set are described in figure legends and were performed using GraphPad Prism 8 software.

## Supporting information

**S1 Fig. Stress granule disassembly rates are not altered in OC43-infected cells.** (A) 293A cells were infected with OC43 at MOI = 1.0 and at the indicated times post-infection mock or OC43-infected cells were treated with 500 μM sodium arsenite (As) for 50 min or left untreated (-). Whole cell lysates were analyzed by western blotting for the levels of eIF2α phosphorylation and the expression levels of the indicated host and viral proteins. (B-D) 293A cells were infected as in A and at 15 hpi mock or OC43-infected cells were treated with 500 μM sodium arsenite (+ As) for 20 or 50 min or left untreated (-). Following 50-min treatment, some cells were washed twice with PBS and provided fresh media to initiate recovery from As-induced stress for the indicated times (Wash-off). The 15 hpi time point was chosen to allow for more infected cells to form SGs for better comparison of assembly and disassembly dynamics vs. mock-infected cells. (B) Immunofluorescence analysis of SG formation in mock infected and OC43-infected cells at the indicated times post-As treatment. (C) Fraction of cells forming SGs was quantified from B. (D) Fraction of cells with SGs from C was normalized to 50 min As treatment timepoint to directly compare assembly and disassembly rates between mock and OC43-infected cells. On all graphs the two-way ANOVA and Tukey multiple comparisons tests were done to determine statistical significance (*, p -value < 0.05; ***, p-value < 0.001; ****, p -value < 0.0001; ns = non-significant). On all plots each data point represents independent biological replicate (N = 3). Error bars = standard deviation, nt = non-treated.
(DOCX)

**S2 Fig. CoV2 Nsp1 increases nuclear TIAR and inhibits TIAR export following transcription inhibition.** Expression of EGFP-tagged CoV2 Nsp1 was induced in 293A[iEGFP-CoV2-Nsp1] cells by treatment with doxycycline (+ Dox) for 24 h. (A) Western blot analysis of cytoplasmic (Cyt.), wash (Wash), and nuclear (Nuc.) fractions of induced and uninduced cells using antibodies for TIAR, cytoplasmic marker β-Tubulin, and nuclear marker Lamin A/C. (B) Nuclear TIAR band intensities from A were quantified and the normalized values to Lamin A/C were plotted (Nuc. lysate, left); ratios of nuclear to cytoplasmic TIAR band intensities were calculated from A (Nuc./Cyt., right). Unpaired Student's t-Tests were done to determine statistical significance (*, p -value < 0.05; ns = non-significant). Each data point represents independent biological replicate (N = 3). Error bars = standard deviation. (C) Immunofluorescence analysis of induced (+ Dox) and uninduced cells, untreated or treated for 4 h with 5 μg/ml actinomycin D (+ ActD). Asterisks highlight nuclei of EGFP-Nsp1 negative cells with depleted TIAR singnal. Arrows indicate EGFP-Nsp1 positive cells with increased nuclear TIAR staining. Scale bar = 50 μm.
(DOCX)

**S3 Fig. CoV2 Nsp1 expression causes nuclear relocalization of PABP.** (A) Immunofluorescence analysis of transiently transfected 293A cells expressing the indicated N-terminally HA-tagged Nsp1 constructs or EGFP control and treated with As showing recruitment of eIF3B to smaller stress granules that form in WT CoV2 Nsp1-expressing cells. (B,C) Immunofluorescence microscopy analysis for subcellular localization of the indicated protein markers in As-treated control 293A[iEGFP-CoV2-Nsp1] cells and cells treated with doxycycline (+ Dox) for 24 h to induce EGFP-Nsp1 expression. (B) Circles outline SGs in EGFP-Nsp1 positive cells that have nuclear PABP and diminished recruitment of PABP to G3BP2-positive SGs. (C) Outsets show normal recruitment of DDX3 and RNase L to EGFP-Nsp1 and G3BP1 double-positive SGs, reduced recruitment of PKR, and lack of recruitment of HSP70 and HSP90A/B in EGFP-Nsp1 expressing cells. Scale bars = 50 μm.
(DOCX)

**S4 Fig. EGFP-G3BP1 overexpressing cells form As-induced SGs and do not form SGs spontaneously.** Immunofluorescence microscopy analysis of 293A[EGFP] and 293A [EGFP-G3BP1] cells untreated (-) or treated with arsenite (+ As) and stained for G3BP2 (magenta). GFP signal is shown in teal. Scale bar = 50 μm.
(DOCX)

## Acknowledgments

We would like to thank Dr. Arinjay Banerjee (Vaccine and Infectious Disease Organization (VIDO), Dr. Karen Mossman (University of McMaster) and Dr. Samira Mubareka (University of Toronto) for the SARS-CoV-2 isolate. We also thank Dr. Craig McCormick (Dalhousie University) for providing NL63 cDNA and Dr. Marta Gaglia (University of Wisconsin-Madison) for providing SARS Nsp1 plasmid. We also thank Dalhousie CORES Flow Cytometry Facility for assistance in generating stable cell lines expressing EGFP-G3BP1 and EGFP. Finally, we thank members of Khaperskyy and Corcoran labs for helpful discussions about experimental design and their critical input on the draft manuscript.

## Author Contributions

**Conceptualization:** Denys A. Khaperskyy.

**Formal analysis:** Stacia M. Dolliver, Mariel Kleer, Maxwell P. Bui-Marinos, Shan Ying, Denys A. Khaperskyy.

**Funding acquisition:** Jennifer A. Corcoran, Denys A. Khaperskyy.

**Investigation:** Stacia M. Dolliver, Mariel Kleer, Maxwell P. Bui-Marinos, Shan Ying, Denys A. Khaperskyy.

**Methodology:** Stacia M. Dolliver, Shan Ying, Denys A. Khaperskyy.

**Project administration:** Jennifer A. Corcoran, Denys A. Khaperskyy.

**Resources:** Denys A. Khaperskyy.

**Supervision:** Jennifer A. Corcoran, Denys A. Khaperskyy.

**Validation:** Stacia M. Dolliver.

**Visualization:** Stacia M. Dolliver, Mariel Kleer, Shan Ying.

**Writing – original draft:** Stacia M. Dolliver, Denys A. Khaperskyy.

**Writing – review & editing:** Stacia M. Dolliver, Mariel Kleer, Jennifer A. Corcoran, Denys A. Khaperskyy.

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
