## [Decision Letter · Decision Letter 0]

18 Nov 2022

Dear Dr. Khaperskyy,

Thank you very much for submitting your manuscript "Nsp1 proteins of human coronaviruses HCoV-OC43 and SARS-CoV2 inhibit stress granule formation" for consideration at PLOS Pathogens. As with all papers reviewed by the journal, your manuscript was reviewed by members of the editorial board and by several independent reviewers. The reviewers appreciated the attention to an important topic. Based on the reviews, we are likely to accept this manuscript for publication, providing that you modify the manuscript according to the review recommendations.

Many thanks for submission of your updated manuscript. It has now been reviewed by 2 of the previous reviewers and I am satisfied that queries from the earlier reviews have been addressed. Reviewer 1 asks for inclusion of data describing the impact of nsp1 within replicating virus. However, I agree with your assessment that generation of recombinant viruses expressing mutated nsp1 is beyond the scope of the current work and appreciate that accessing these viruses from other research groups may not be practical.

Please see below some minor text edits for clarity:

1. Fig 2C - clarify that the quantitation is from sil. treated cells

2. Line 620 - modify 7A to 7B

Sincerely,

Helena Jane Maier, DPhil

Guest Editor

PLOS Pathogens

Volker Thiel

Section Editor

PLOS Pathogens

Kasturi Haldar

Editor-in-Chief

PLOS Pathogens

orcid.org/0000-0001-5065-158X

Michael Malim

Editor-in-Chief

PLOS Pathogens

orcid.org/0000-0002-7699-2064

Many thanks for submission of your updated manuscript. It has now been reviewed by 2 of the previous reviewers and I am satisfied that queries from the earlier reviews have been addressed. Reviewer 1 asks for inclusion of data describing the impact of nsp1 within replicating virus. However, I agree with your assessment that generation of recombinant viruses expressing mutated nsp1 is beyond the scope of the current work and appreciate that accessing these viruses from other research groups may not be practical.

Please see below some minor text edits for clarity:

1. Fig 2C - clarify that the quantitation is from sil. treated cells

2. Line 620 - modify 7A to 7B

Reviewer Comments (if any, and for reference):

Reviewer's Responses to Questions

**Part I - Summary**

Reviewer #1: The authors studied inhibition of SG formation in HCoV-OC43-infected cells, SARS-CoV-2-infected cells, cells expressing N proteins, and those expressing nsp1 of both HCoVs.

Reviewer #2: This reviewer would like to thank the authors for provided additional results that addressed all the points raised in the previous review of their manuscript. All the points are convincingly answered and the manuscript is now worthy of publication in P.Path.

**Part II – Major Issues: Key Experiments Required for Acceptance**

Reviewer #1: Although the editor and the reviewers pointed out that the lack of experiments examining the role of nsp1 for suppressing SG formation in infected cells represented a notable weakness of this study, the revised manuscript did not include studies using SARS-CoV-2 mutant encoding a biologically inactive nsp1. As such a SARS-CoV-2 mutant has been generated and is available to others according to ref 46, testing the role of nsp1 for suppression of SG formation in SARS-CoV-2-infected cells would be feasible without generating a new mutant SARS-CoV-2 in the authors’ lab. Inclusion of the data using mutant SARS-CoV-2 would significantly improve significance of this study.

Reviewer #2: NA

**Part III – Minor Issues: Editorial and Data Presentation Modifications**

Reviewer #1: 1. Line 386-387. The authors still stated that both OC43 and CoV2 nsp1 proteins block translation initiation. Are there any papers showing inhibition of translation initiation by OC43 nsp1? If not, this section should be altered. The data shown in Fig 5 did not prove that OC43 nsp1 suppresses translation initiation, as OC43 nsp1 may suppress another step(s) in translation or may affect protein stability.

2. Fig. 5I. Do 99A and 125A show statistically significant reduction of SG as compared with EGFP?

3. Lines 461-463. This sentence should be rewritten as the meaning is not very clear.

Reviewer #2: NA

PLOS authors have the option to publish the peer review history of their article (what does this mean?). If published, this will include your full peer review and any attached files.

Reviewer #1: No

Reviewer #2: **Yes: **Nicolas Locker

Figure Files:

Data Requirements:

Reproducibility:

References:

---

## [Editor Report · Decision Letter 1]

3 Dec 2022

Dear Dr. Khaperskyy,

We are pleased to inform you that your manuscript 'Nsp1 proteins of human coronaviruses HCoV-OC43 and SARS-CoV2 inhibit stress granule formation' has been provisionally accepted for publication in PLOS Pathogens.

Best regards,

Helena Jane Maier, DPhil

Guest Editor

PLOS Pathogens

Volker Thiel

Section Editor

PLOS Pathogens

Kasturi Haldar

Editor-in-Chief

PLOS Pathogens

orcid.org/0000-0001-5065-158X

Michael Malim

Editor-in-Chief

PLOS Pathogens

orcid.org/0000-0002-7699-2064
---

## [Editor Report · Acceptance letter]

7 Dec 2022

Dear Dr. Khaperskyy,

We are delighted to inform you that your manuscript, "Nsp1 proteins of human coronaviruses HCoV-OC43 and SARS-CoV2 inhibit stress granule formation," has been formally accepted for publication in PLOS Pathogens.

Best regards,

Kasturi Haldar

Editor-in-Chief

PLOS Pathogens

orcid.org/0000-0001-5065-158X

Michael Malim

Editor-in-Chief

PLOS Pathogens

orcid.org/0000-0002-7699-2064